# APP and APLP2 interact with the synaptic release machinery and facilitate transmitter release at hippocampal synapses

Tomas Fanutza[1†], Dolores Del Prete[1†], Michael J Ford[2], Pablo E Castillo[3*], Luciano D'Adamio[1*]

[1]Department of Microbiology and Immunology, Albert Einstein College of Medicine, New York, United States; [2]MS Bioworks, LLC, Ann Arbor, United States; [3]Dominick P. Purpura Department of Neuroscience, Albert Einstein College of Medicine, New York, United States

**Abstract** The amyloid precursor protein (APP), whose mutations cause familial Alzheimer's disease, interacts with the synaptic release machinery, suggesting a role in neurotransmission. Here we mapped this interaction to the $NH_2$-terminal region of the APP intracellular domain. A peptide encompassing this binding domain -named JCasp- is naturally produced by a $\gamma$-secretase/caspase double-cut of APP. JCasp interferes with the APP-presynaptic proteins interaction and, if linked to a cell-penetrating peptide, reduces glutamate release in acute hippocampal slices from wild-type but not APP deficient mice, indicating that JCasp inhibits APP function.The APP-like protein-2 (APLP2) also binds the synaptic release machinery. Deletion of APP and APLP2 produces synaptic deficits similar to those caused by JCasp. Our data support the notion that APP and APLP2 facilitate transmitter release, likely through the interaction with the neurotransmitter release machinery. Given the link of APP to Alzheimer's disease, alterations of this synaptic role of APP could contribute to dementia.

*For correspondence: pablo.castillo@einstein.yu.edu (PEC); luciano.dadamio@einstein.yu.edu (LD'Adamio)

[†]These authors contributed equally to this work

Competing interests: The authors declare that no competing interests exist.

## Introduction

Human genetic evidence and studies in *App*-knock-out (KO) mice indicate that the amyloid precursor protein (APP) plays an important physiological role in the central nervous system. In humans, a polymorphism in *APP* that reduces APP processing protects from sporadic Alzheimer's disease (AD) and normal aging-dependent cognitive decline (*De Strooper and Voet, 2012*, *Jonsson et al., 2012*). In contrast, mutations in *APP* and in genes that regulate APP processing – such as *PSENs* and *BRI2/ITM2B* – cause familial AD (FAD) and the AD-like familial British dementia and familial Danish dementia (*De Strooper, 2007*, *De Strooper et al., 2010*, *De Strooper and Voet, 2012*, *Garringer et al., 2010*, *Matsuda et al., 2005*, *Giliberto et al., 2008*, *Matsuda et al., 2009*, *Tanzi, 2012*, *Vidal et al., 1999*, *2000*). Analysis of *App*-KO mice has also suggested a synaptic role for APP (*Korte et al., 2012*, *Marcello et al., 2012*, *Octave et al., 2013*, *Randall et al., 2010*, *Turner et al., 2003*, *Venkitaramani et al., 2007*). For example, hippocampal *App*-KO neurons in culture show enhanced α-amino-3-hydroxy-5-methyl-4-isoxazolepropionic acid (AMPA) and N-methyl-D-aspartate (NMDA) receptor-mediated synaptic transmission and increased size of the readily releasable synaptic vesicle pool (RRP). In addition, long-term potentiation (LTP), a form of synaptic plasticity that underlies memory formation, is impaired in *App*-KO mice; and these LTP deficits are clearly noticeable in

**eLife digest** Alzheimer's disease has been linked to mutations in a gene encoding the amyloid precursor protein (APP). Mutations in this gene cause early onset Alzheimer's disease in some families. Studies in animals suggest this mutant form of the protein may interfere with the messages sent between brain cells. But it remains unclear how this protein works even when it isn't mutated.

One reason it has been difficult to figure out how APP works is because it is a long protein and is often cut into smaller, but still functional proteins. It is unclear which parts of the protein are involved in sending messages between brain cells, but previous research has allowed scientists to zero in on one stretch of the protein containing just 50 amino acids.

Now, Fanutza, Del Prete et al. reveal that this short stretch of APP interacts with the proteins that control the release of glutamate, one of the brain's most important excitatory neurotransmitter. This stretch is cut from the full protein and released into the cell as a protein fragment (or peptide). Further experiments showed that this naturally occurring peptide interferes with the activity of APP, and in doing so reduces the release of glutamate from brain cells in mice. However, there was no effect on glutamate release when this peptide was introduced inside brain cells from mutant mice that lacked APP.

Together, the results show that this 50-amino acid stretch of APP promotes glutamate release between brain cells, and that a peptide containing the same stretch of the protein interferes with this process. Altered glutamate neurotransmission could explain the memory loss and thinking problems seen in early Alzheimer's disease. Therefore, drug treatments that aim to restore normal glutamate transmission may prove a beneficial therapeutic strategy in Alzheimer's disease.

aged but not in juvenile *App*-KO mice (*Dawson et al., 1999*, *Korte et al., 2012*, *Priller et al., 2006*, *Venkitaramani et al., 2007*).

Although the evidence indicating that APP plays a role in several processes controlling synaptic function is strong, the molecular mechanisms are poorly understood. A confounding factor when studying APP function is the complex APP metabolism. The best-characterized cascade involves a sequential β/γ-secretase cleavage of APP. In this pathway, APP is cleaved by β-secretase/BACE1 into a soluble ectodomain (sAPPβ) and a membrane-bound COOH-terminal fragment of 99 amino acids (C99 or β C-terminal fragment [CTF]). C99/β-CTF is cleaved by the γ-secretase to produce Aβ peptides, which are believed to be the main culprit of AD, and the APP intracellular domain (AID/AICD). Alternatively, α-secretase cleaves APP in the Aβ sequence into sAPPα and a membrane-bound COOH-terminal fragment of 83 amino acids (C83 or α-CTF). C83/α-CTF is cleaved by the γ-secretase to produce a short peptide comprising the COOH-terminus of Aβ, called P3, and AID/AICD (*Annaert et al., 2000*, *De Strooper, 2003*, *Selkoe and Kopan, 2003*, *Sisodia and St George-Hyslop, 2002*, *Vassar et al., 1999*). Another processing pathway, potentially important yet less studied, involves cleavage of APP in the cytoplasmic domain – between $D^{664}$ and $A^{665}$, following the numbering of the main brain APP isoform of 695 amino acids – by caspase-6, -3, and -8. Caspases can in theory process all APP metabolites comprising this intracellular motif, such as full-length APP, C99/β-CTF, C83/α-CTF, and AID/AICD. So far, cleavages of full-length APP and AID/AICD by caspases have been described (*Bertrand et al., 2001*, *Gervais et al., 1999*, *Lu et al., 2000*, *Madeira et al., 2005*, *Pellegrini et al., 1999*, *Weidemann et al., 1999*). Processing of full-length APP generates a membrane bound $NH_2$-terminal fragment called APP-Ncas and a COOH-terminal intracellular peptide called APP-Ccas or C31 (*Bertrand et al., 2001*, *Gervais et al., 1999*, *Lu et al., 2000*, *Madeira et al., 2005*, *Pellegrini et al., 1999*, *Weidemann et al., 1999*). Caspase cleavage of AID/AICD splits the cytoplasmic domain of APP into two soluble cytosolic peptides: the aforementioned Ccas/C31 and JCasp. The latter comprises the $NH_2$-terminal peptide segment of the APP intracellular domain (*Bertrand et al., 2001*, *Madeira et al., 2005*). Recently, a new secretase activity (called η-secretase) has been described (*Willem et al., 2015*). Cleavage by η-secretase occurs primarily at amino acids 504–505 of APP695 and releases a ~500 amino acid-long ecto-domain and a membrane-bound fragment called CTF-η. Interestingly, CTF-η is further processed by α- and β-secretases to release a long and a short Aη peptide (Aη-α and Aη-β, respectively).

Many of the APP metabolites produced by these cleavages are biologically active. AID/AICD modulates programmed cell death, gene transcription and $Ca^{2+}$ homeostasis (*Baek et al., 2002*, *Cao and Südhof, 2001*, *Checler et al., 2007*, *Cupers et al., 2001*, *Hamid et al., 2007*, *Kim et al., 2003*, *Leissring et al., 2002*, *Liu et al., 2007*, *Madeira et al., 2005*, *Pardossi-Piquard et al., 2005*, *Passer et al., 2000*, *von Rotz et al., 2004*). The caspase-derived APP fragments Ccas/C31 and JCasp (*Bertrand et al., 2001*, *Lu et al., 2000*, *Madeira et al., 2005*) also possess toxic activities. Aβ both regulates normal synaptic transmission and prompts deficits in synaptic plasticity (*Fogel et al., 2014*, *Mota et al., 2014*, *Puzzo and Arancio, 2013*, *Puzzo et al., 2008*, *Shankar et al., 2007*, *Sivanesan et al., 2013*, *Um et al., 2012*). sAPPα has neurotrophic functions and is involved in synaptic plasticity (*Hick et al., 2015*, *Milosch et al., 2014*). sAPPβ promotes axon pruning and neurodegeneration (*Nikolaev et al., 2009*). Aη-α can reduce LTP (*Willem et al., 2015*). β-CTF mediates long-term synaptic plasticity deficits, behavioral deficits and neurodegeneration (*Neve et al., 1996*, *Neve and Robakis, 1998*, *Oster-Granite et al., 1996*, *Tamayev et al., 2012b*, *2012c*).

The presence of the functionally redundant APP-like protein-1 and -2 (APLP1 and APLP2) is another confounding factor. APLP1 and APLP2 are processed like APP (*Heber et al., 2000*, *Herms et al., 2004*, *Müller and Zheng, 2012*, *Scheinfeld et al., 2002*, *von Koch et al., 1997*) and release intracellular peptides – called APLP2-intracellular domain (ALID) 1 and ALID2, respectively – that, like AID/AICD, can potentially regulate transcription (*Scheinfeld et al., 2002*, *2003*). The evidence that *Aplp2*-KO and *App*-KO mice are viable, whereas combined *App/Aplp2*-dKO mice develop neuromuscular junction deficits and die shortly after birth (*Wang et al., 2005*) vividly illustrates the functional redundancy of APP and APLP2. The robust phenotype of *App/Aplp2*-dKO mice can be used to map the functional domains of APP by expressing *App* knock-in mutations on an *Aplp2*-KO genetic background. This in vivo functional mapping has identified an essential role for the highly conserved APP intracellular domain in the patterning of neuromuscular junction and survival (*Barbagallo et al., 2011a*, *2011b*, *Li et al., 2010*), indicating that the intracellular region of APP is functionally important in vivo

The APP cytoplasmic domain is very short (~50 amino acids) and does not possess recognizable enzymatic domains, raising the possibility that APP functions could be dictated by molecular interaction of the APP cytoplasmic domain with intracellular proteins. Using an unbiased proteomic approach, we found that the brain APP intracellular domain-interactome includes synaptic vesicles trans-membrane proteins, proteins associated with synaptic vesicles and the presynaptic active zone (*Del Prete et al., 2014*). Many of these proteins, such as *cis*- and *trans*-SNAREs, complexin, and the $Ca^{2+}$ sensors synaptotagmins, function as key regulators of synaptic vesicles exocytosis (*McMahon and Boucrot, 2011*, *Rizzoli, 2014*, *Royle and Lagnado, 2010*, *Südhof, 2012*, *2013*, *Südhof and Rizo, 2011*, *Xu et al., 2009*). Remarkably a similar brain APP–presynaptic interactome was identified in transgenic mice overexpressing human APP (*Kohli et al., 2012*, *Norstrom et al., 2010*). These findings are consistent with the evidence that APP in the brain is found at presynaptic terminals and synaptic vesicles (*Del Prete et al., 2014*, *Groemer et al., 2011*, *Laßek et al., 2014*, *Wilhelm et al., 2014*, *Yang et al., 2005*).

The fact that APP is present at synaptic vesicles and the presynaptic active zone – specialized region of the presynaptic membrane where neurotransmitter release occurs – and that the APP intracellular domain interacts with the neurotransmitter release machinery suggests that APP regulates transmitter release. Here, we investigated this proposition. By recording neuronal activity in hippocampal brain slices while using and APP-derived peptide that interferes with the APP-neurotransmitter release machinery interaction and acts as a dominant negative inhibitor of APP, we implicate APP in the physiological regulation of the synaptic release of glutamate at hippocampal excitatory synapses. Moreover, we found that APLP2 can compensate for loss of APP function and that genetic ablation of both APP and APLP2 protein expression produces glutamatergic synaptic transmission deficits similar to those caused by JCasp, confirming that JCasp acts as a dominant negative inhibitor of some functions of APP in wild-type (WT) animals. Our results indicate that APP and APLP2 fine-tune synaptic vesicle release likely through the interaction with, and regulation of, the molecular machinery that controls synaptic vesicle exocytosis.

## Results

### APP interacts with presynaptic proteins via the NH$_2$-terminal portion of its intracellular domain

Given that the intracellular domain of APP interacts with proteins that mediate synaptic vesicle exocytosis (*Del Prete et al., 2014*), APP could regulate transmitter release. To test this hypothesis, one should target those functions associated with these presynaptic APP interactions, possibly without directly interfering in the function of other APP domains/regions, such as Aβ or the long APP extracellular domain and derived products (sAPPα and sAPPβ). As an initial step toward the generation of biological tools to test the hypothesis, we mapped the APP intracellular domain that interacts with the exocytosis machinery, and used a proteomic approach to identify the brain proteins that interact with the N-terminus (JCasp) and C-terminus (Ccas) sub-domains of the APP intracellular region. JCasp and Ccas are two naturally occurring fragments that are generated by a double γ-secretase/caspases cleavage of APP (*Figure 1A and B*) (*Bertrand et al., 2001*, *Gervais et al., 1999*, *Lu et al., 2000*, *Madeira et al., 2005*, *Pellegrini et al., 1999*, *Weidemann et al., 1999*). It is therefore conceivable that this physiological proteolytic event has been selected to separate two functionally distinct intracellular regions of APP. In line with this thought, presynaptic interactors of APP bind to the NH$_2$-terminal intracellular region JCasp. Other APP-binding proteins, such as the APP interactor Ddb1 (*Watanabe et al., 1999*) and the Ddb1-interacting proteins Cul4A and Cul4B (*O'Connell and Harper, 2007*), specifically bind the COOH-terminal APP intracellular region Ccas (*Table 1*, Source Data were uploaded to the PRIDE repository – European Bioinformatics Institute (EBI), Cambridge, UK – with Proteome Xchange identifier #48018), underscoring the reliability of the mapping method. In addition, JCasp, but not ScJCasp, which contains the same amino acids of JCasp but in a scrambled order, competes for the binding of Syp, Vamp2, and Syt2 to St-AID, supporting the notion that JCasp contains the binding domain of APP for these presynaptic proteins (*Figure 1B and C*).

These interactions suggest a presynaptic function of APP in vivo. This hypothesis is substantiated by a number of studies showing localization of APP at presynaptic terminals and synaptic vesicles. Biochemical studies have shown that endogenous APP is present in fractions highly enriched in synaptic vesicles and the active zone (*Del Prete et al., 2014*, *Groemer et al., 2011*, *Laßek et al., 2013*, *Lundgren et al., 2015*). Immune-precipitation of APP from mouse brains showed interaction of APP with key regulators of synaptic vesicles exocytosis. Although the possibility that these interactions happen post-cellular lysis cannot be formally discarded, these data are consistent with a presynaptic localization of APP in vivo. Immuno-electron microscopy experiments (performed using the specific Y-188 anti-APP antibody, see below) have indicated that APP is localized at presynaptic terminals and in synaptic vesicles in the hippocampus (*Del Prete et al., 2014*). Finally, localization of APP at presynaptic terminals has been reported also in primary neuronal cultures where it can regulate synaptic transmission functioning as a presynaptic receptor for Aβ (*Fogel et al., 2014*).

The data discussed above support a presynaptic function of APP. Next, we analyzed the pattern of expression of APP in the hippocampus, a brain structure critically involved in memory formation. We revisited this question in view of a recent report showing that most of the antibodies that have been used to determine the localization of endogenous APP are non-specific as they label *App*-KO and WT cells/tissues in a similar fashion (*Guo et al., 2012*). The authors showed that Y188 was the only APP-specific antibody. The evidence that Y188 does not stain *App*-KO hippocampi confirmed the specificity of this antibody (*Figure 1E and F*). In addition, low magnification analysis shows that APP is expressed, albeit at different levels, in all hippocampal circuits, including the CA1 neurons and the stratum radiatum containing the CA3 projections (Schaffer collaterals, or SC) to CA1 neurons (*Figure 1E*). Higher magnification analysis confirms that APP staining is stronger in CA1 pyramidal neurons as compared to the stratum radiatum (*Figure 1F*). Several factors may explain these differences: 1) APP is synthesized and undergoes maturation in the endoplasmic reticulum (ER) and Golgi that are highly concentrated in cell bodies; 2) APP may be expressed at synapses at levels that are, albeit functionally important, too low to be clearly detected by Y188; 3) Y188 recognizes the C-terminal epitope of APP, and this epitope may be functionally engaged in multi-molecular interactions at presynaptic terminals and may therefore be scarcely accessible to Y188. Nevertheless, the staining shown in *Figure 1F* indicates that APP is indeed expressed in the stratum radiatum, where it partially co-localizes with the presynaptic protein Bassoon.

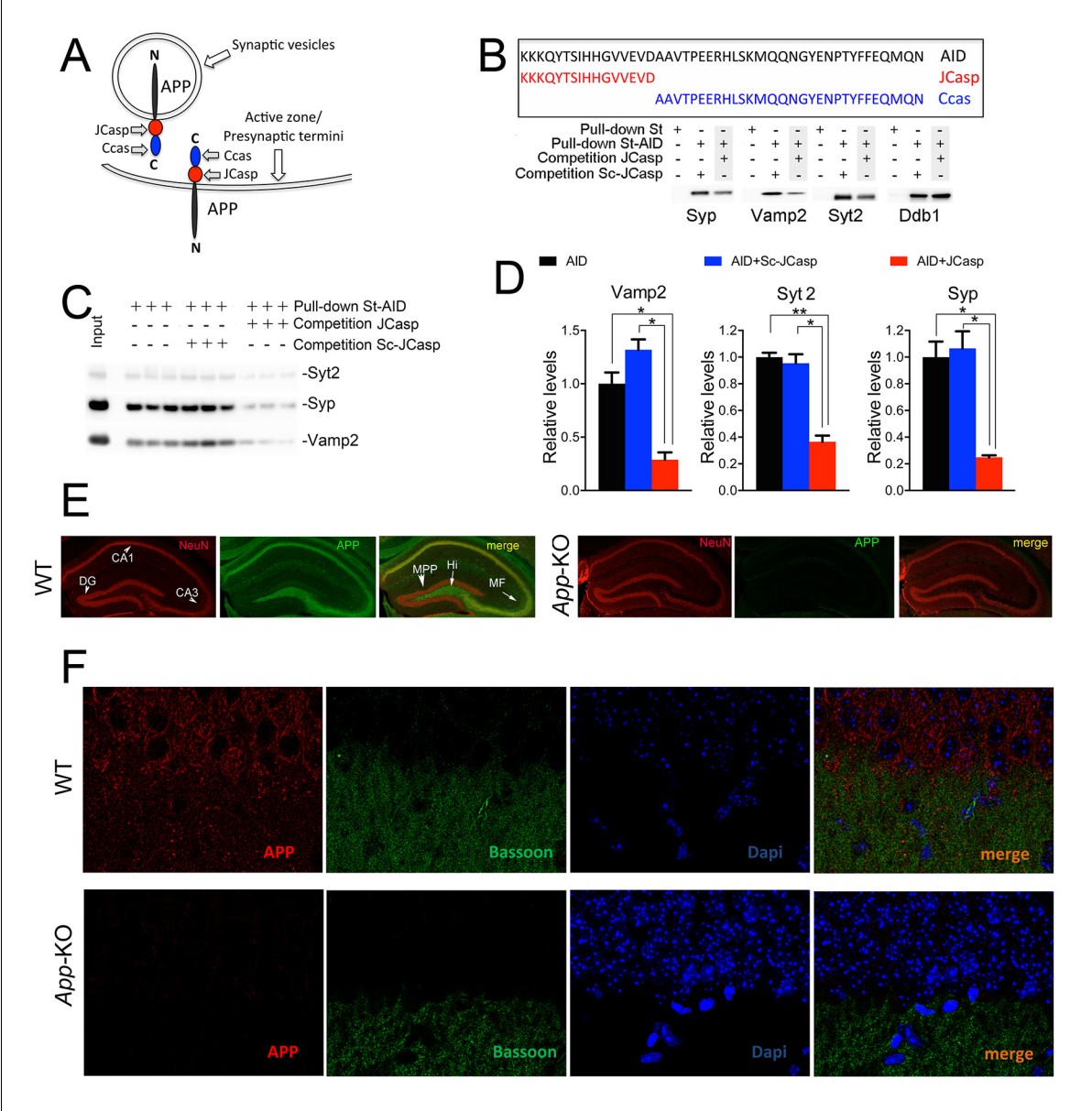

**Figure 1.** The AID–presynaptic interactome binds to JCasp. (A) The APP intracellular domain, composed by the two sub-regions JCasp and Ccas, is always cytosolic whether APP molecules are localized at synaptic vesicles or the active zone. (B) Sequences of AID, JCasp, and Ccas used as baits in the proteomic experiments. (C) Brain lysates were incubated for 30 min with 10 μM of either JCasp or ScJCasp. After incubation, the lysates were affinity purified on StrepTactin columns bound to the indicated St-peptides. Bound proteins were analyzed by Western blot. JCasp reduces binding of Syp, Vamp2, and Syt2, but not Ddb1, to St-AID. (D) Triplicate experiment showing that JCasp, but not ScJCasp, significantly reduces binding of Syp, Vamp2, and Syt2 to St-AID. (E) Quantification of the data shown in (D). (F) Hippocampal slices from 8-week-old WT or *App*-KO mice were stained with an αNeuN (red, left panels) and an αAPP (green middle panel) antibody. The images were merged in the right panels. APP is widely distributed in the hippocampus. The strongest staining is seen in the Hi, MF, CA3, and CA1. The staining for APP is specific since there is no signal in *App*-KO hippocampi. (G) Staining for APP (red, left panels) and Bassoon (green middle panel) show that APP is expressed in the *Stratum radiatum* where it partially co-localizes with Bassoon (arrows in the merged image, right panel). Again, the staining for APP and the co-localization spots are specific as shown by their absence in *App*-KO hippocampi. AID, APP intracellular domain; APP, amyloid precursor protein; WT, wild type.

## Intracellular delivery of JCasp decreases excitatory synaptic transmission

This presynaptic APP-interacting network can stem from APP molecules localized at synaptic vesicles, presynaptic membranes or both, and it is likely formed via the direct interaction of the

**Table 1.** Proteins that bind JCasp are in red/bold. Proteins that bind Ccas are in blue/the other entries. Source Data containing all search results, coverage maps, peptide lists and product ion data were uploaded to the PRIDE repository (European Bioinformatics Institute (EBI), Cambridge, UK) with Proteome Xchange identifier #48018. Details of protein identification data related to the proteins discussed in this manuscript can be found in Table1. The table file contains: the list of proteins identified (Column 1); the database accession numbers (Column 2); the spectral counts (SpC, Columns 3–7); Spectral Abundance Factor (SAF, Columns 8–12); subsequent Normalized Spectral Abundance Factor (NSAF, Columns 13–17). This was based on the equation: NSAF = (SpC/MW)/Σ(SpC/MW)N, where SpC = Spectral Counts, MW = Protein MW in kDa and N = Total Number of Proteins, NSAF contains NSAF values without alone.

| Prot. | Acc. Num. | SpC | | | | | SAF | | | | | NSAF | | | | |
| --- | --- | --- | --- | --- | --- | --- | --- | --- | --- | --- | --- | --- | --- | --- | --- | --- |
| | | ST | ST-AID | ST | ST-Ccas | ST-JCasp | ST | ST-AID | ST | ST-Ccas | ST-JCasp | ST | ST-AID | ST | ST-Ccas | ST-JCasp |
| **Ddb1** | **spQ3U1J4\|** | 0 | 418 | 0 | 730 | 0 | 0.0000 | 3.2913 | 0.0000 | 5.7480 | 0.0000 | 0.0000 | 0.0139 | 0.0000 | 0.0366 | 0.0000 |
| Syt1 | sp\|P46096\| | 0 | 88 | 0 | 0 | 113 | 0.0000 | 1.8723 | 0.0000 | 0.0000 | 2.4043 | 0.0000 | 0.0079 | 0.0000 | 0.0000 | 0.0129 |
| Stxbp1 | sp\|O08599\| | 0 | 44 | 0 | 8 | 46 | 0.0000 | 0.6471 | 0.0000 | 0.1176 | 0.6765 | 0.0000 | 0.0027 | 0.0000 | 0.0007 | 0.0036 |
| **Cul4a** | **sp\|Q3TCH7\|** | 0 | 40 | 0 | 36 | 0 | 0.0000 | 0.4545 | 0.0000 | 0.4091 | 0.0000 | 0.0000 | 0.0019 | 0.0000 | 0.0026 | 0.0000 |
| Sv2b | sp\|Q8BG39\| | 0 | 55 | 0 | 0 | 66 | 0.0000 | 0.7143 | 0.0000 | 0.0000 | 0.8571 | 0.0000 | 0.0030 | 0.0000 | 0.0000 | 0.0046 |
| Syngr3 | sp\|Q8R191\| | 0 | 40 | 0 | 6 | 79 | 0.0000 | 1.6000 | 0.0000 | 0.2400 | 3.1600 | 0.0000 | 0.0068 | 0.0000 | 0.0015 | 0.0170 |
| **Cul4b** | **sp\|A2A432\|** | 0 | 19 | 0 | 15 | 0 | 0.0000 | 0.1712 | 0.0000 | 0.1351 | 0.0000 | 0.0000 | 0.0007 | 0.0000 | 0.0009 | 0.0000 |
| Stx1b | sp\|P61264\| | 0 | 23 | 0 | 0 | 31 | 0.0000 | 0.6970 | 0.0000 | 0.0000 | 0.9394 | 0.0000 | 0.0029 | 0.0000 | 0.0000 | 0.0050 |
| Vamp2 | sp\|P63044\| | 0 | 23 | 0 | 0 | 46 | 0.0000 | 1.7692 | 0.0000 | 0.0000 | 3.5385 | 0.0000 | 0.0075 | 0.0000 | 0.0000 | 0.0190 |
| Syngr1 | sp\|O55100\| | 0 | 8 | 0 | 0 | 30 | 0.0000 | 0.3077 | 0.0000 | 0.0000 | 1.1538 | 0.0000 | 0.0013 | 0.0000 | 0.0000 | 0.0062 |
| Sv2a | sp\|Q9JIS5\| | 0 | 11 | 0 | 1 | 23 | 0.0000 | 0.1325 | 0.0000 | 0.0120 | 0.2771 | 0.0000 | 0.0006 | 0.0000 | 0.0001 | 0.0015 |
| Snap25 | sp\|P60879\| | 0 | 16 | 0 | 12 | 35 | 0.0000 | 0.6957 | 0.0000 | 0.5217 | 1.5217 | 0.0000 | 0.0029 | 0.0000 | 0.0033 | 0.0082 |
| Stx1a | sp\|O35526\| | 0 | 2 | 0 | 0 | 21 | 0.0000 | 0.0606 | 0.0000 | 0.0000 | 0.6364 | 0.0000 | 0.0003 | 0.0000 | 0.0000 | 0.0034 |
| Stx7 | sp\|O70439\| | 0 | 3 | 0 | 0 | 4 | 0.0000 | 0.1000 | 0.0000 | 0.0000 | 0.1333 | 0.0000 | 0.0004 | 0.0000 | 0.0000 | 0.0007 |
| Syt2 | sp\|P46097\| | 0 | 15 | 0 | 0 | 36 | 0.0000 | 0.3191 | 0.0000 | 0.0000 | 0.7660 | 0.0000 | 0.0013 | 0.0000 | 0.0000 | 0.0041 |
| Syt12 | sp\|Q920N7\| | 0 | 3 | 0 | 0 | 10 | 0.0000 | 0.0638 | 0.0000 | 0.0000 | 0.2128 | 0.0000 | 0.0003 | 0.0000 | 0.0000 | 0.0011 |
| Stx12 | sp\|Q9ER00\| | 0 | 6 | 0 | 0 | 5 | 0.0000 | 0.1935 | 0.0000 | 0.0000 | 0.1613 | 0.0000 | 0.0008 | 0.0000 | 0.0000 | 0.0009 |

intracellular domain of APP with some release machinery protein. The other components are most likely indirectly associated to APP (*Figure 2A*). Since JCasp interferes with these interactions in vitro, if delivered inside neurons, JCasp could hamper the function of this presynaptic APP–interacting complex in vivo (*Figure 2B*) and reveal the function of this intracellular APP domain.

Thus, we systematically characterized the effect of JCasp on excitatory synaptic transmission at SC–CA1 pyramidal cell synapses. To provide intracellular delivery, we synthesized JCasp and the negative control peptide ScJCasp linked to the cell-penetrating peptide (*Davidson et al., 2004*) Penetratin1 (Pen1) at the N-terminus (*Figure 2B*), a strategy that we have previously used to deliver peptides to murine hippocampal neurons in vivo (*Tamayev et al., 2012c*). First, we verified whether

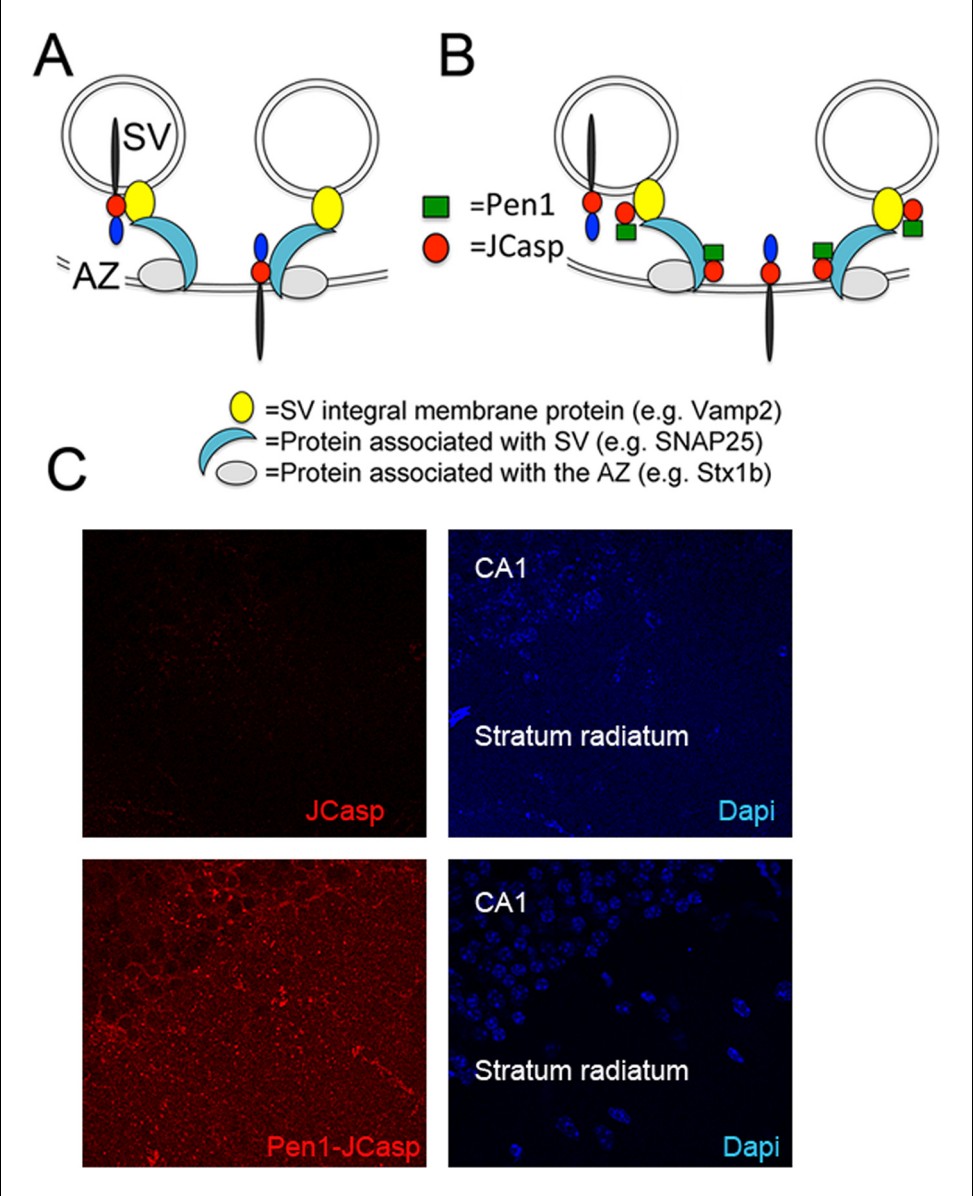

**Figure 2.** Pen1-JCasp is delivered intra-neuronally. Models of assembly of the amyloid precursor protein (APP) presynaptic interactome and mechanism of action of Pen1-JCasp. (**A**) SV = synaptic vesicles; AZ = active zone. SV proteins and proteins associated to SV and to the AZ bind via direct and indirect interactions with the JCasp region of APP molecules localized to either SV or the AZ. The interactions depicted here are hypothetical. (**B**) Pen1-JCasp is delivered intracellularly and can interfere with the interaction of APP with presynaptic proteins. (**C**) Immune-fluorescence experiment showing that Pen1 delivers the JCasp peptide in hippocampal neurons.

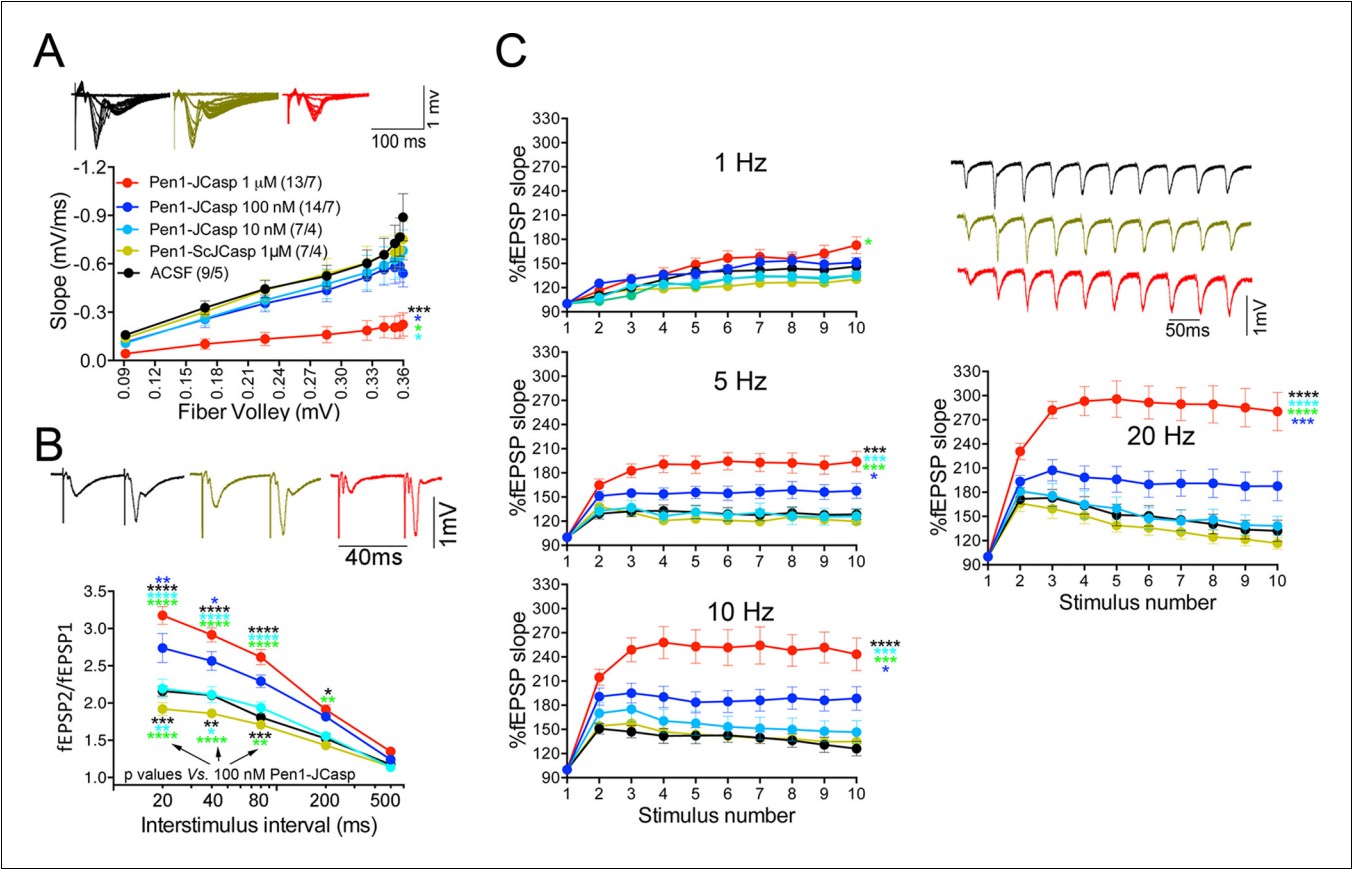

**Figure 3.** Pen1-JCasp impairs synaptic plasticity. (**A**) CA1 recordings of hippocampal slices incubated with either artificial cerebrospinal fluid (ACSF), 1 µM Pen1-ScJCasp or Pen1-JCasp at the indicated concentrations. The synaptic input/output (I/O) relationship was obtained by plotting the fiber volley amplitude against the initial slope of the evoked field excitatory postsynaptic potential (fEPSP). Representative traces are shown on top. The slope values of I/O recording for each group were compared for statistical assessments. It has been found that 1 µM Pen1-JCasp significantly reduces synaptic transmission as compared to ACSF (*), Pen1-ScJCasp (*), 10 nM (*), and 100 nM Pen1-JCasp (*p). Representative traces are shown. (**B**) Average paired-pulse facilitation (PPF) (2nd fEPSP/1st fEPSP) plotted as a function of the inter-stimulus interval. Representative traces of fEPSPs evoked at 40 ms inter-stimulus interval are shown. Pen1-JCasp increases PPF. (**C**) Synaptic facilitation elicited by stimulus trains at 1, 5, 10, and 20 Hz. fEPSP slopes are normalized to the slope of the first fEPSP of the stimulus train. Representative traces of fEPSPs evoked at 20 Hz are shown. Stimulus artifacts are removed for clarity. Pen1-JCasp increases frequency facilitation (FF) in a frequency and dose-dependent manner. Statistical assessments were performed by: one-way analysis of variance (ANOVA) followed by Tukey's multiple comparisons test for slopes of I/O curves; two-way repeated measures (RM) ANOVA followed by Tukey's multiple comparisons test for PPF and FF (*p < 0.05; **p < 0.01; ***p < 0.001; ****p < 0.0001). In FF, 10 nM Pen1-JCasp increased facilitation in a statistically significant manner only at the 10th stimulation (indicated with (*) and (*)). The number of recordings and the number of mice analyzed for each group are shown in (**C**). All data represent means ± SEM. The complete statistical analyses are shown in the attached Excel file.

The following source data is available for figure 3:

**Source data 1.** Source data for statistical analysis of input/output curves in *Figure 3A*.
**Source data 2.** Source data for statistical analysis of paired-pulse facilitation in *Figure 3B*.
**Source data 3.** Source data for statistical analysis of frequency facilitation at 1 Hz in *Figure 3C*.
**Source data 4.** Source data for statistical analysis of frequency facilitation at 5 Hz in *Figure 3C*.
**Source data 5.** Source data for statistical analysis of frequency facilitation at 10 Hz in *Figure 3C*.
**Source data 6.** Source data for statistical analysis of frequency facilitation at 20 Hz in *Figure 3C*.

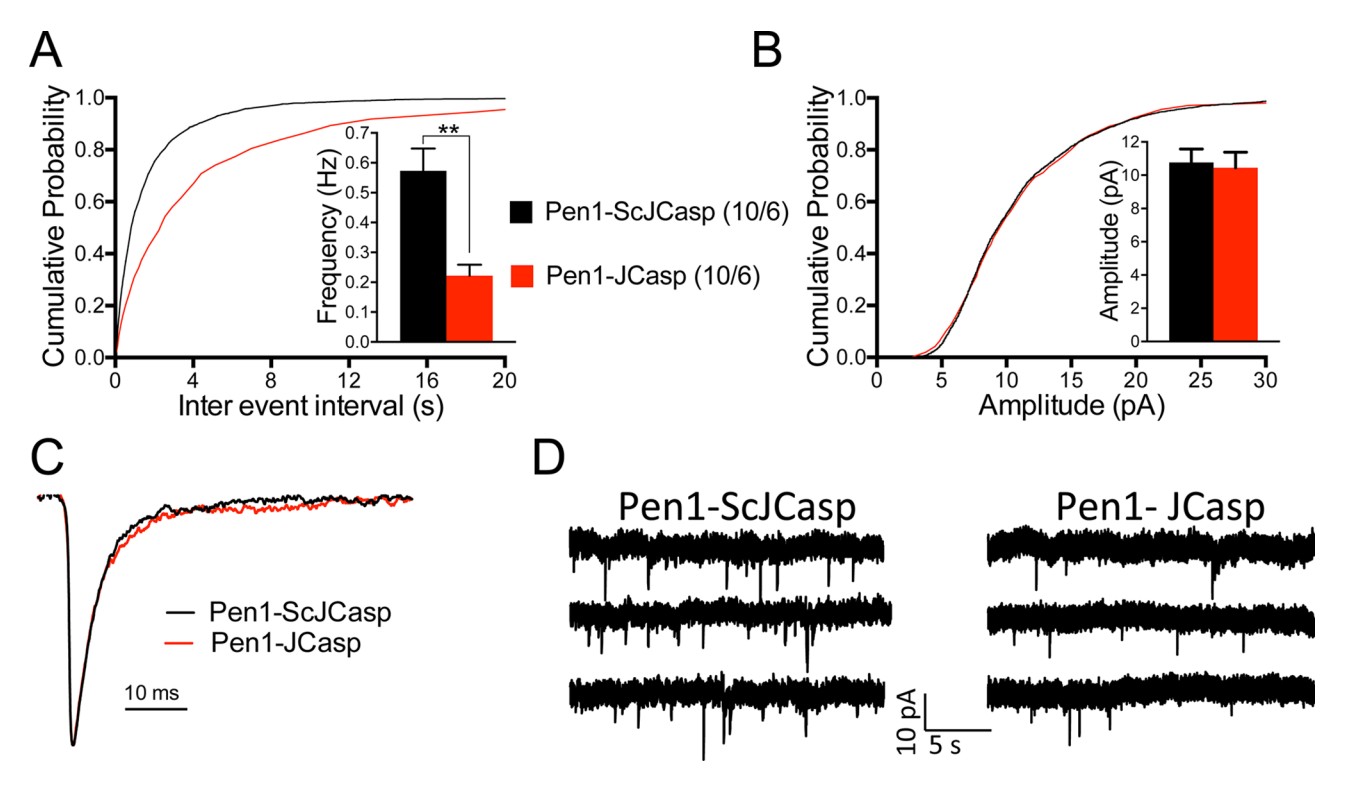

**Figure 4.** Pen1-JCasp reduces frequency of excitatory miniature currents. (A) Cumulative probability of α-amino-3-hydroxy-5-methyl-4-isoxazolepropionic acid receptor (AMPAR) mediated miniature excitatory postsynaptic current (EPSC) (mEPSC) inter-event intervals. (B) Cumulative probability of AMPAR-mediated mEPSC amplitudes. Insets in cumulative probability graphs represent average mEPSC frequency (A) and amplitudes (B). mEPSC frequency was significantly reduced by Pen1-JCasp. (C) Average mEPSC of the two groups were not significantly different. (D) Representative recording traces of miniature EPSCs are shown. The number of recordings and the number of mice analyzed for each group are shown in (A). All data represent means ± SEM. Statistical assessments were performed by paired t test (**p < 0.01).

The following source data is available for figure 4:

**Source data 1.** Source data for statistical analysis of mEPSCs frequency in *Figure 4A*.

Pen1-JCasp efficiently delivers JCasp at hippocampal SC–CA1 synapses. To this end, we incubated hippocampal slices with 1 µM of either biotinylated JCasp or biotinylated Pen1-JCasp. Later, 5 hr after incubation, slices were stained for the peptide using Streptavidin. As shown in *Figure 2C*, Pen1-JCasp, but not JCasp, is detected in neurons of the CA1 pyramidal neurons and in the stratum radiatum, which contains the SCs. Thus, we determined whether Pen1-JCasp alters synaptic transmission at SC–CA1 synapses. The experiments were performed as follows: soon after preparation the brain slices were incubated in 200 ml of artificial cerebrospinal fluid (ACSF) containing the indicated concentrations of either Pen1-JCasp or the negative control peptide Pen1-ScJCasp. Recordings were performed between ~5 and 8 hr after preparation of slices and incubation with Pen1 peptides. We found that Pen1-JCasp impaired basal synaptic transmission, as indicated by a reduction in the input/output (I/O) function (*Figure 3A*). These effects were dose-dependent and specific since the negative control peptide Pen1-ScJCasp did not alter synaptic transmission.

To test whether the reduction in I/O function could be due to presynaptic impairment, we first estimated the probability of release (Pr) by analyzing two forms of short-term synaptic plasticity: paired-pulse facilitation (PPF) and synaptic frequency facilitation (FF) in response to a stimulus train (10 stimuli at either 1, 5, 10, or 20 Hz). These forms of short-term synaptic plasticity are determined, at least in part, by changes in Pr, such that a decrease in Pr leads to an increase in facilitation and vice versa (*Zucker and Regehr, 2002*). Pen1-JCasp but not Pen1-ScJCasp significantly increased both PPF (*Figure 3B*) and FF (*Figure 3C*) in a dose-dependent manner. In addition, Pen1-JCasp

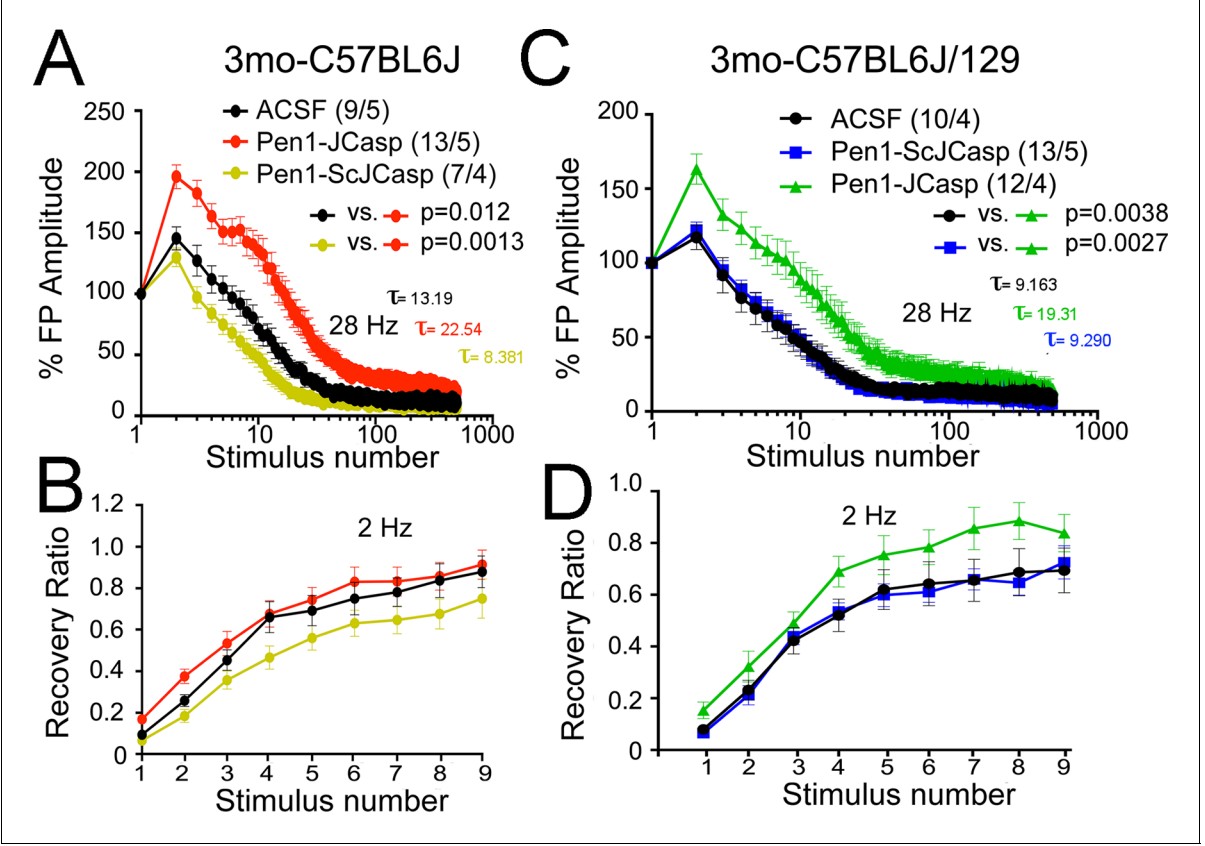

**Figure 5.** Pen1-JCasp reduces the rate of depletion of glutamatergic vesicles. (**A**) Artificial cerebrospinal fluid (ACSF) vs Pen1-JCasp, p < 0.05; Pen1-SCJCasp vs. Pen1-JCasp, p < 0.01; (**B**) ACSF vs. Pen1-JCasp, p< 0.01; Pen1-SCJCasp vs. Pen1-JCasp, p < 0.01. Recovery kinetics (**C** and **D**) are not affected. The experiments were performed in 3-month-old C57Bl/6j (**A** and **C**) and C57Bl/6j-129 hybrid mice (**B** and **D**). All experiments shown in this paper were performed in C57Bl/6j mice, except for the experiments shown in *Figure 7*; *App*-KO and wild-type (WT) littermate animals were on a C57Bl/6j-129 hybrid genetic background. Data represent means ± SEM. Statistical assessments were performed by nonlinear regression curve fit, one phase decay.

The following source data is available for figure 5:

**Source data 1.** Source data for statistical analysis of decay rate shown in *Figure 5A*.

**Source data 2.** Source data for statistical analysis of decay rate shown in *Figure 5B*.

reduced miniature excitatory postsynaptic current (EPSC) (mEPSC) frequency (*Figure 4A*), but had no effect on mEPSC amplitude (*Figure 4B*) or waveform (*Figure 4C*), suggesting a presynaptic mechanism of action. We also delivered a high-frequency train of stimuli to deplete the RRP. The rate of RRP depletion, as measured by the suppression of synaptic transmission during repetitive stimulation, is proportional to the initial P$r$ (i.e., a reduction in P$r$ would decrease this rate and vice versa, an increase in P$r$ would accelerate it). Indeed, Pen1-JCasp slowed down the rate of synaptic suppression induced by a 500 stimuli, 28 Hz train (*Figure 5A and B*). After depletion, the recycling of synaptic vesicles replenishes the RRP. We found that the recovery phase after depletion – measured by stimulating at 2 Hz – was unaffected by Pen1-JCasp (*Figure 5C and D*), arguing against an alteration of synaptic vesicle recycling as a cause of the reduced synaptic strength. Taken together, our findings strongly suggest that Pen1-JCasp reduces basal synaptic transmission by interfering with glutamate release.

Given that several APP metabolites have been involved in a wide range of neuronal functions, including synaptic plasticity, Pen1-JCasp could reduce synaptic transmission indirectly by altering APP processing. To address this possibility, we tested the effect of Pen1-JCasp on APP processing.

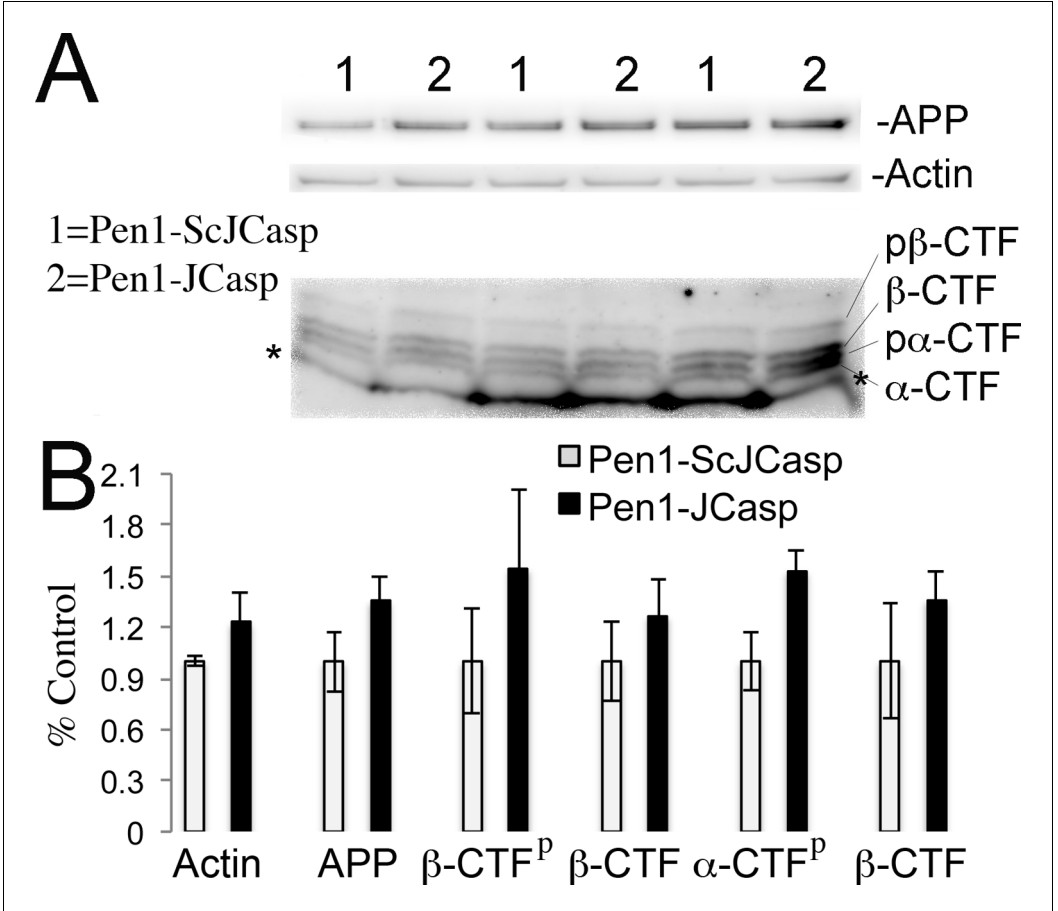

**Figure 6.** Pen1-JCasp does not alter amyloid precursor protein (APP) processing. (**A**) Western blot analysis of Actin, APP, β C-terminal fragment (CTF) and α-CTF in hippocampal slices incubated for 8 hr with either 1 µM Pen1-JCasp or 1 µM Pen1-ScJCasp. β-CTF$^p$ and α-CTF$^p$ are phosphorylated forms of β-CTF and α-CTF, respectively. (**B**) Quantification of the data: for each protein, the average signal obtained with 1 µM Pen1-ScJCasp is arbitrarily considered 1; the average signal obtained with 1 µM Pen1-JCasp is expressed as a percent of the control. The * indicates an aspecific band, which is also present in *App*-KO lysates (not shown).

Several soluble APP metabolites resulting from APP processing, such as APPβ, sAPPα, and Aβ, are diluted into the large volume (200 ml) of ACSF used for culturing hippocampal slices during recordings. Therefore, it is difficult to accurately measure the levels of these soluble APP metabolites. However, other APP metabolites, that is, β-CTF and α-CTF, remain membrane bound and can be measured by Western blot, together with full-length APP. Pen1-JCasp does not change levels of full-length APP, β-CTF, and α-CTF (*Figure 6A and B*), arguing against the possibility that Pen1-JCasp alters synaptic plasticity via modulation of APP processing.

Based on our findings, we hypothesized that JCasp reduces transmitter release by interfering with the intracellular interactions of APP with key regulators of vesicle exocytosis. To test this hypothesis we compared the effects of Pen1-JCasp and JCasp side-by-side . Pen1-JCasp, but not JCasp, which lacks the cell penetrating properties of Pen1 and is not detected inside neurons (*Figure 2C*), reduced basal synaptic transmission (*Figure 7A*) and increased PPF (*Figure 7B*) and FF (*Figure 7C*). Thus, to suppress transmitter release, the JCasp peptide must access the intracellular compartment.

A recent report shows that Aβ in cultured neurons triggers APP dimerization that, by interacting via the intracellular domain with $G\alpha_o\beta_1\gamma_2$ protein, increases $Ca^{+2}$ influx through $Ca_v2$ channels, thereby enhancing P*r* (*Fogel et al., 2014*). Thus, Pen1-JCasp could reduce P*r* by inhibiting this pathway and reducing $Ca^{+2}$ influx. However, we found that Pen1-JCasp was equally effective in the

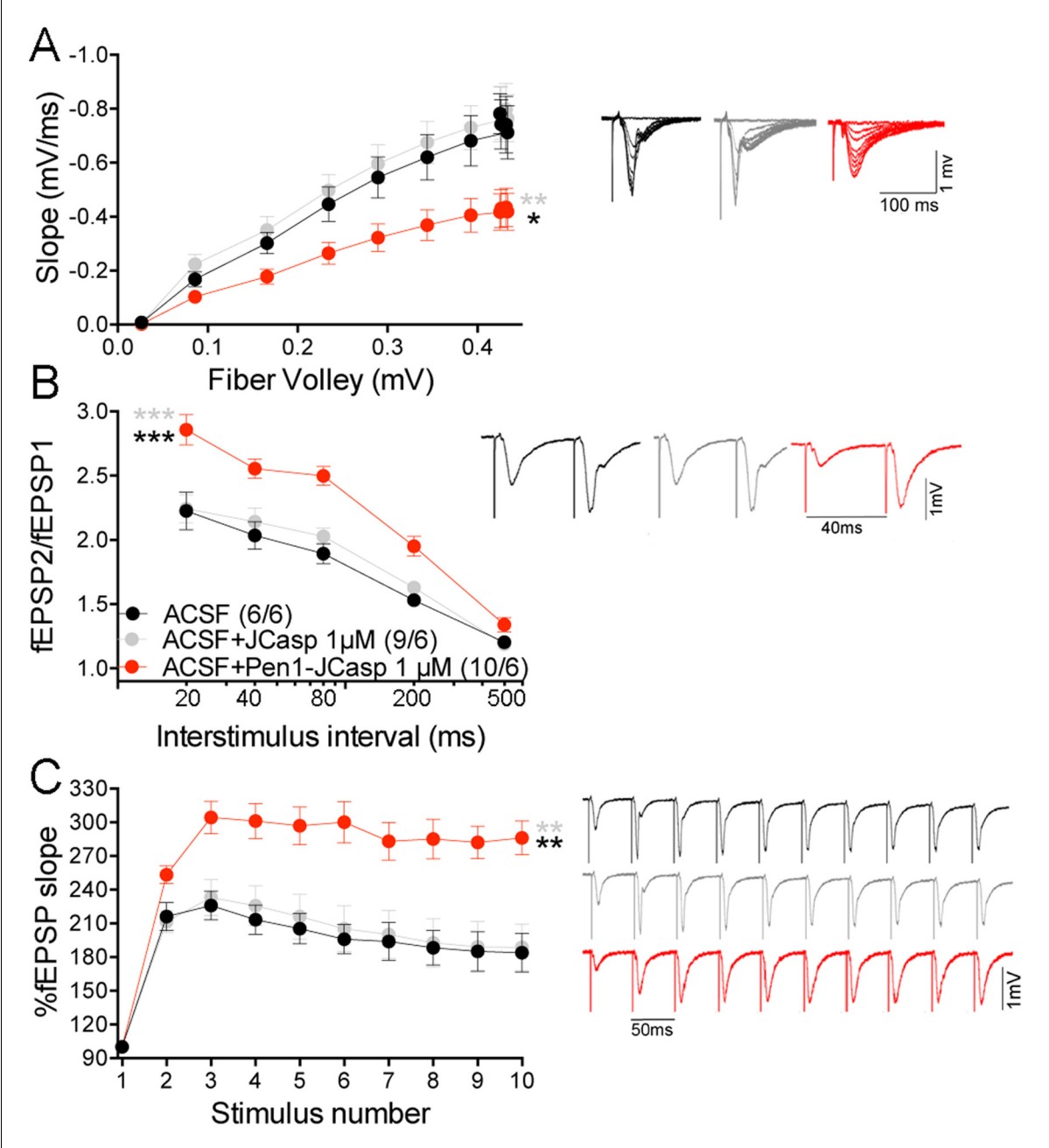

**Figure 7.** Pen1-JCasp impairs excitatory synapses via an intracellular mechanism. Pen1-JCasp, but not JCasp, reduces basal synaptic transmission (**A**) paired-pulse facilitation (PPF) (**B**) and frequency facilitation (FF) at 20 Hz (**C**). Representative traces are shown. Stimulus artifacts are removed from the FF traces for clarity. The number of recordings and of mice analyzed are shown in (**B**). Statistical assessments were performed by: one-way analysis of variance (ANOVA) followed by Tukey's multiple comparisons test for slopes of input/output (I/O) curves; two-way repeated measures (RM) ANOVA followed by Tukey's multiple comparisons test for PPF and FF. *Pen1-JCasp 1 µM vs. artificial cerebrospinal fluid (ACSF), *Pen1-JCasp 1 µM vs. JCasp; *p < 0.05; **p < 0.01; ***p < 0.001; ****p < 0.0001. The complete statistical analyses are shown in the attached Excel file. All data represent means ± SEM.

The following source data is available for figure 7:

**Source data 1.** Source data for statistical analysis of input/output curves in *Figure 7A*.

**Source data 2.** Source data for statistical analysis of paired-pulse facilitation in *Figure 7B*.

**Source data 3.** Source data for statistical analysis of frequency facilitation at 20 Hz in *Figure 7C*.

*Figure 7 continued*

presence of 100 µM $Cd^{+2}$, a manipulation that non-selectively blocks $Ca^{+2}$ influx through voltage-gated $Ca^{+2}$ channels (*Figure 8*).

## JCasp inhibits the function of endogenous APP

Next, we determined whether JCasp has any effect on SC–CA1 synaptic transmission in *App*-KO and WT littermates. Interestingly, Pen1-JCasp reduced I/O function (*Figure 9A*) and increased PPF (*Figure 9B*) and FF (*Figure 9C*) in WT but not in *App*-KO animals, suggesting that Pen1-JCasp does not alter transmission at APP-lacking synapses.

The fact that Pen1-JCasp-mediated suppression of synaptic transmission requires APP expression not only indicates that Pen1-JCasp acts as a dominant negative inhibitor of endogenous APP but also suggests that endogenous APP regulates glutamate release at excitatory synapses. However, we found no differences in synaptic facilitation between *App*-KO and WT hippocampi (*Figure 9B and C*), suggesting that compensatory mechanisms operate in *App*-KO mice to counterbalance the loss of APP function. Good candidates for these redundant functions are APLP1 and APLP2, the other two members of the APP protein family; particularly APLP2 since functional redundancy

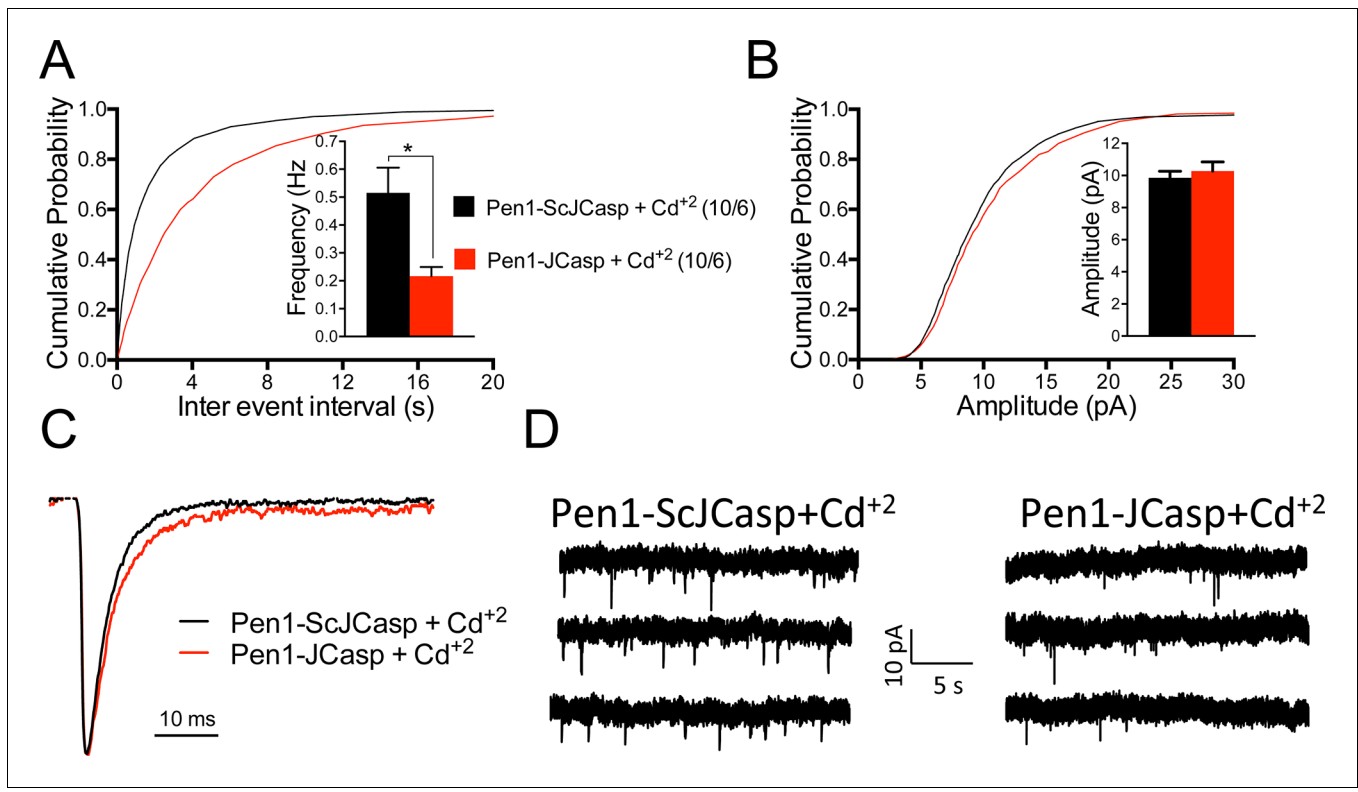

**Figure 8.** Pen1-JCasp reduces the frequency of miniature excitatory postsynaptic currents (EPSCs) (mEPSCs) independently of alterations of $Ca^{+2}$ influx. (**A**) Cumulative probability of α-amino-3-hydroxy-5-methyl-4-isoxazolepropionic acid receptor (AMPAR) mediated mEPSC inter-event intervals. (**B**) Cumulative probability of AMPAR-mediated mEPSC amplitudes. Insets in cumulative probability graphs represent average mEPSC frequency (**A**) and amplitudes (**B**). mEPSC frequency was significantly reduced by Pen1-JCasp in the presence of $Cd^{+2}$. (**C**) Average mEPSC of the two groups were not significantly different. (**D**) Representative recording traces of miniature EPSCs are shown. The number of recordings and the number of mice analyzed for each group are shown in (**A**). All data represent means ± SEM. Statistical assessments were performed by paired t test (*p < 0.05).

The following source data is available for figure 8:

**Source data 1.** Source data for statistical analysis of mEPSCs frequency in *Figure 8A*.

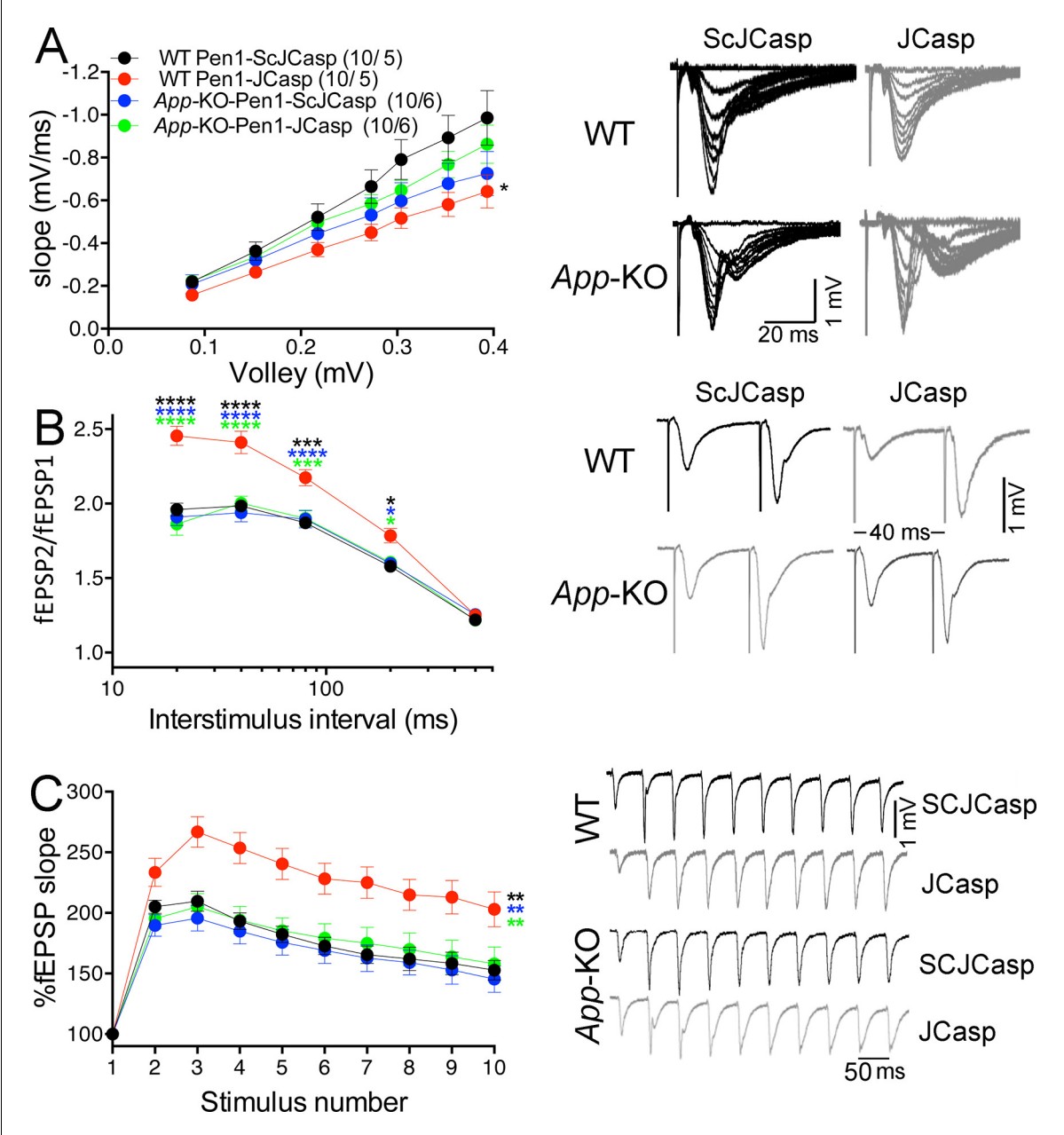

**Figure 9.** Amyloid precursor protein (APP) is required for the inhibitory effect of Pen1-JCasp on synaptic transmission. (A) Analysis of slopes of each input/output (I/O) recording by one-way analysis of variance (ANOVA) followed by Tukey's multiple comparisons test showed no significant differences among the groups. However, uncorrected Fisher's LSD test shows that Pen1-JCasp reduces basal synaptic transmission in wild-type (WT) hippocampal slices as compared to WT slices treated with Pen1-ScJCasp. Pen1-JCasp increases paired-pulse facilitation (PPF) (B) and frequency facilitation (FF) (C) in WT but not *App*-KO excitatory Schaffer collateral (SC) synapses. Representative traces are shown on the right of summary plots. The number of recordings and of mice analyzed for each group are shown in (A). All data represent means ± SEM. Statistical assessment by two-way repeated measures (RM) ANOVA shows significant differences in PPF and FF between WT mice treated with Pen1-JCasp and the other three experimental groups (*, vs. WT+Pen1-ScJCasp; *, vs. *App*-KO+Pen1-ScJCasp; *, vs. *App*-KO+Pen1-JCasp; *p < 0.05; **p < 0.01; ***p < 0.001; ****p < 0.0001). The complete statistical analyses are shown in the attached Excel file.

The following source data is available for figure 9:

**Source data 1.** Source data for statistical analysis of input/output curves in *Figure 9A*.

**Source data 2.** Source data for statistical analysis of paired-pulse facilitation in *Figure 9B*.

*Figure 9 continued on next page*

*Figure 9 continued*

**Source data 3.** Source data for statistical analysis of frequency facilitation at 20 Hz in *Figure 9C*.

between APP and APLP2 has already been demonstrated (*Barbagallo et al., 2011b*, *Dawson et al., 1999*, *von Koch et al., 1997*, *Wang et al., 2005*). Of note, APLP1 and APLP2 have also been localized to presynaptic terminals (*Laßek et al., 2013*), further supporting a compensatory function at presynaptic terminals.

## Loss of APP and APLP2 reduces excitatory synaptic transmission

As a first step to determine whether APLP2 shares this presynaptic function of APP, we characterized the brain interactome of the ALID2 using the same unbiased proteomic approach used for Ccas and JCasp. We synthesized two peptides: control Strep-tag only peptide and St-ALID2. Interestingly, the brain interactome of ALID2 comprises most of the key proteins of the presynaptic release machinery that bind to JCasp and AID (*Table 2*, Source Data were uploaded to the PRIDE repository – European Bioinformatics Institute (EBI), Cambridge, UK – with Proteome Xchange identifier #58725).

The preceding data are consistent with the idea that APLP2, via its intracellular domain, can compensate the function of the cytoplasmic domain of APP in *App*-KO synapses. Yet, the evidence that Pen1-JCasp does not alter synaptic transmission in *App*-KO synapses (*Figure 9*) indicates that Pen1-JCasp acts as an inhibitor of the intracellular domain of APP but not APLP2 in vivo. Consistent with this possibility JCasp and the corresponding APLP2 region (**RKRQYGTISHGIVEVD**) are only 63% identical, and JCasp does not interfere with the interaction of ALID2 with Vamp2, Syt2, and Syp (*Figure 10*). Thus, Pen1-Jcasp is a tool to selectively study the function of APP.

To directly test whether APLP2 can compensate for loss of APP function, we monitored the effect of constitutive APP, APLP2, and APP+APLP2 deletion on excitatory synaptic transmission at SC–CA1 synapses in acute brain slices. Because *App/Aplp2*-dKO mice die within the first 4 weeks of age, for comparison reasons all experiments were performed on 16- to 24-day-old mice of both sexes. Animals were on a mixed C57BL/6J and 129 genetic background. *Aplp2*-KO and *App/Aplp2*-dKO showed reduced basal synaptic transmission as compared to both WT and *App*-KO mice; however, these differences reached statistical significance only when compared to *App*-KO samples (*Figure 11A*).

**Table 2.** Interaction of ALID2 with presynaptic protein. Source Data containing all search results, coverage maps, peptide lists and product ion data were uploaded to the PRIDE repository (European Bioinformatics Institute (EBI), Cambridge, UK) with Proteome Xchange identifier #58725. The table file contains: the list of proteins identified (Column 1); the database accession numbers (Column 2); the spectral counts (SpC, Columns 3-7); Spectral Abundance Factor (SAF, Columns 8-12); subsequent Normalized Spectral Abundance Factor (NSAF, Columns 13-17). This was based on the equation: NSAF = (SpC/MW)/$\Sigma$(SpC/MW)N, where SpC = Spectral Counts, MW = Protein MW in kDa and N = Total Number of Proteins, NSAF contains NSAF values without alone.

| Prot. | Acc. Num. | ST SpC | ST-ALID2 SpC | ST SAF | ST-ALID2 SpC | ST NSAF | ST-ALID2 SpC |
|---|---|---|---|---|---|---|---|
| Syt1 | sp\|P46096\| | 0 | 29 | 0.0000 | 0.3235 | 0.0000 | 0.0045 |
| Stxbp1 | sp\|O08599\| | 0 | 22 | 0.0000 | 0.6471 | 0.0000 | 0.0027 |
| Sv2b | sp\|Q8BG39\| | 0 | 15 | 0.0000 | 0.1948 | 0.0000 | 0.0034 |
| Syngr3 | sp\|Q8R191\| | 0 | 7 | 0.0000 | 0.28 | 0.0000 | 0.0067 |
| Stx1b | sp\|P61264\| | 0 | 6 | 0.0000 | 0.1212 | 0.0000 | 0.0029 |
| Vamp2 | sp\|P63044\| | 0 | 11 | 0.0000 | 0.8461 | 0.0000 | 0.0203 |
| Syngr1 | sp\|O55100\| | 0 | 6 | 0.0000 | 0.8846 | 0.0000 | 0.0055 |
| Sv2a | sp\|Q9JIS5\| | 0 | 14 | 0.0000 | 0.4019 | 0.0000 | 0.0023 |
| Syt2 | sp\|P46097\| | 0 | 18 | 0.0000 | 0.3829 | 0.0000 | 0.0053 |
| Syph | sp\|Q62277\| | 0 | 8 | 0.0000 | 0.2353 | 0.0000 | 0.0049 |

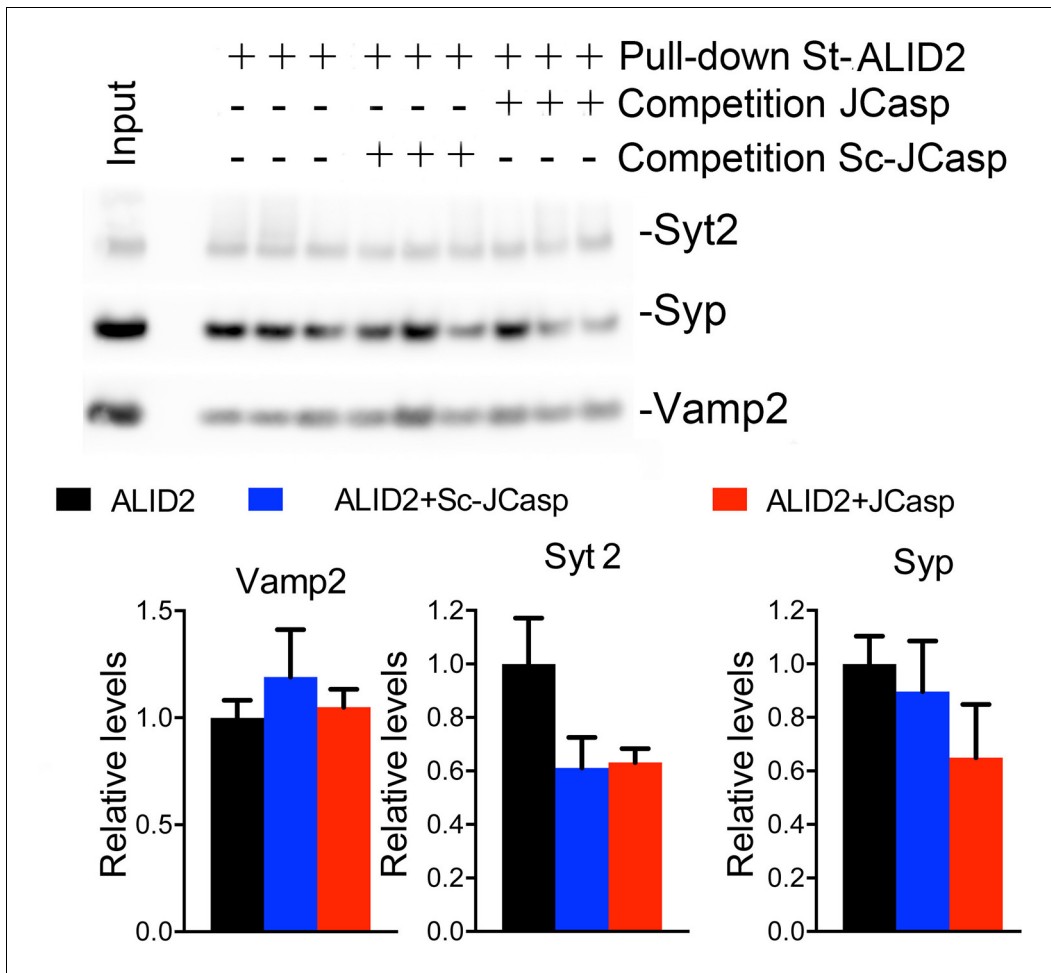

**Figure 10.** JCasp does not compete for the interaction of amyloid precursor protein (APP) like protein-2 (APLP2) intracellular domain (ALID2) with Syt2, Syp, and Vamp2. Brain lysates were incubated for 30 min with 10 μM of either JCasp or ScJCasp. After incubation, the lysates were affinity purified on StrepTactin columns bound to the indicated St-peptides. Bound proteins were analyzed by Western blot. JCasp does not reduce binding of Syp, Vamp2, and Syt2 to St-ALID2. The graphs at the bottom of the figure show quantitative analysis of the Western blot data.

The *App/Aplp2*-dKO mice showed an increase in PPF (*Figure 11B*) and FF (*Figure 11C*), suggesting a reduction in P*r* (*Zucker and Regehr, 2002*). The magnitude PPF and FF in *App*-KO and *Aplp2*-KO SC–CA1 synapses was not significantly different than in WT mice (*Figure 11B and C*).

Next we monitored mEPSCs. Interestingly, we found that mEPSC frequency was decreased by genetic ablation of APP and APLP2 expression, with APLP2 deletion having a greater effect as compared to APP deletion (*Figure 12A*, WT>*App*-KO>*Aplp2*-KO>*App/Aplp2*-dKO). However, these differences reached statistical significance only between WT *Vs. Aplp2*-KO and WT *Vs. App/Aplp2*-dKO samples. Thus, our results showing changes in short-term plasticity and mEPSC activity strongly suggest that while APP and APLP2 have redundant effects on glutamate release, APLP2 compensates for loss of APP function in *App*-KO mice more efficiently than APP compensates loss of APLP2 function in *Aplp2*-KO hippocampi. Remarkably, the synaptic deficits observed in *App/Aplp2*-dKO SC–CA1 excitatory synapses are strikingly similar to those induced by JCasp in WT hippocampi, supporting the notion that a common mechanism underlies these deficits.

We also found that the mEPSC decay time was increased in *App/Aplp2*-dKO as compared to WT, *App*-KO, and *Aplp2*-KO (*Figure 12C, D*), suggesting that APP and APLP2 may possess some redundant postsynaptic function, consistent with previous observations indicating a postsynaptic role for some APP metabolites, such as Aβ (*Gu et al., 2009*, *Hsieh et al., 2006*). Interestingly, Pen1-JCasp

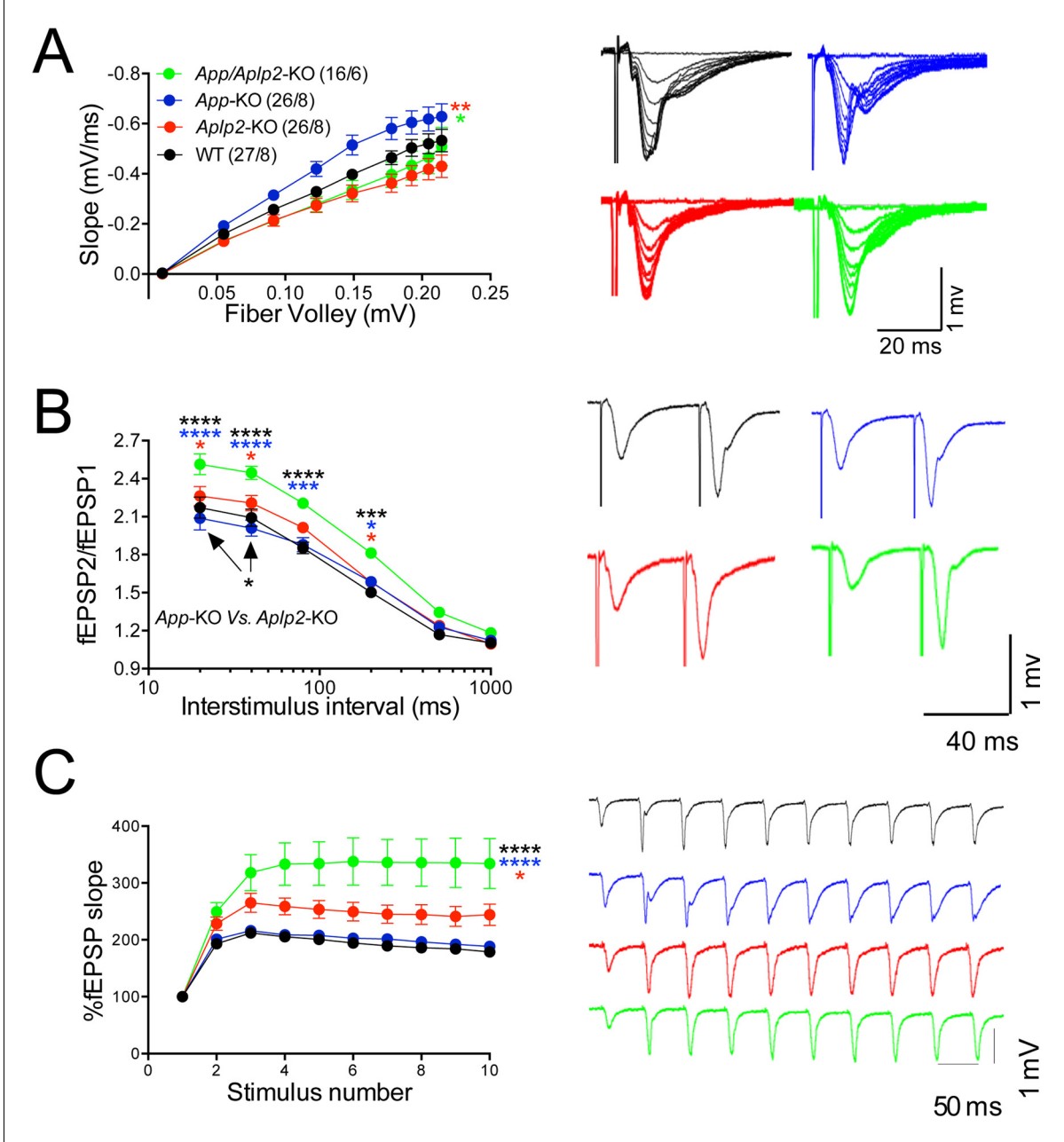

**Figure 11.** Amyloid precursor protein (APP) like protein-2 (APLP2) compensates for loss of APP function. (**A**) Input/output (I/O) recording showed significant differences between different genotypes (*App*-KO vs. *Aplp2*-KO, **p < 0.01; *App*-KO vs. *App/Aplp2*-dKO, *p < 0.05). PPR (**B**) and frequency facilitation (FF) (**C**) are significantly increased in *App/Aplp2*-dKO as compared to all other genotypes. Representative traces are shown on the right of summary plots. The number of recordings and of mice analyzed for each group are shown in (**A**). Statistical assessment by two-way repeated measures analysis of variance (RM ANOVA) followed by Tukey's multiple comparisons tests. All data represent means ± SEM (*p < 0.05; **p < 0.01; ***p < 0.001).

The following source data is available for figure 11:

**Source data 1.** Source data for statistical analysis of input/output curves in *Figure 11A*.

**Source data 2.** Source data for statistical analysis of paired-pulse facilitation in *Figure 11B*.

**Source data 3.** Source data for statistical analysis of frequency facilitation at 20Hz in *Figure 11C*.

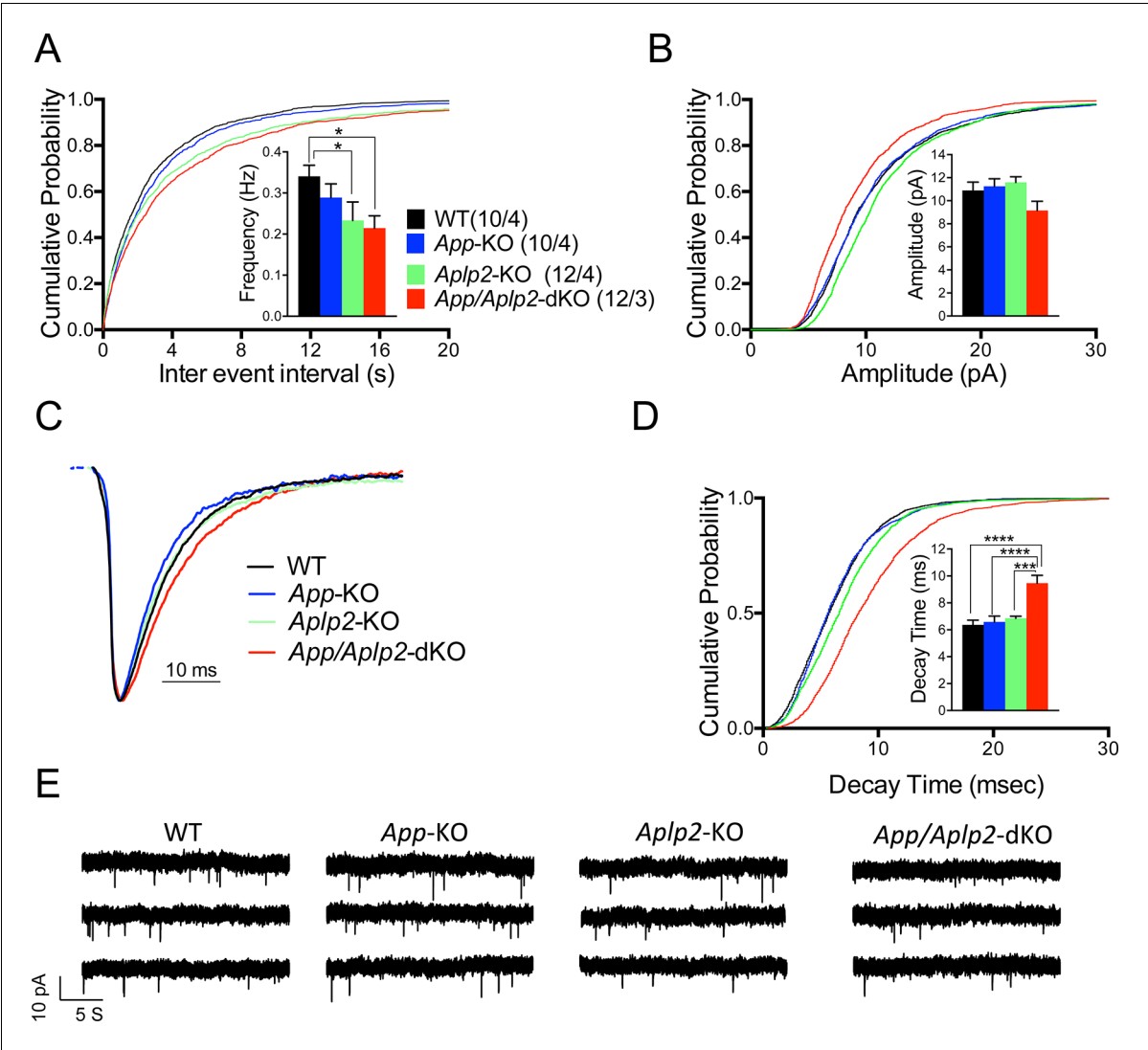

**Figure 12.** Reduced frequency of miniature excitatory postsynaptic currents (EPSCs) (mEPSCs) in Aplp2-KO and App/Aplp2-dKO CA1 pyramidal neurons. (A) Cumulative probability of α-amino-3-hydroxy-5-methyl-4-isoxazolepropionic acid receptor (AMPAR)-mediated mEPSC inter-event intervals. (B) Cumulative probability of AMPAR-mediated mEPSC amplitudes. Insets in cumulative probability graphs represent average mEPSC frequency (A) and amplitudes (B). mEPSC frequency was significantly reduced in *Aplp2*-KO and *App/Aplp2*-dKO CA1 pyramidal neurons as compared to wild-type (WT) but not *App*-KO littermates. (C) Average mEPSC of the four groups were not significantly different. (D) Cumulative probability of AMPAR-mediated mEPSC decay time with insets representing average mEPSC decay time. mEPSC decay time was significantly increased in *App/Aplp2*-dKO CA1 pyramidal neurons as compared to WT, *App*-KO, and *Aplp2*-KO littermates. (E) Representative recording traces of miniature EPSCs are shown. The number of recordings and the number of mice analyzed for each group are shown in (A). All data represent means ± SEM. Statistical assessment was performed using ordinary one-way ANOVA followed by uncorrected Fisher's least significant difference (LSD) multiple comparisons test (*p < 0.05; **p < 0.01; ***p < 0.001; ****p < 0.0001).

The following source data is available for figure 12:

**Source data 1.** Source data for statistical analysis of mEPSCs frequency in *Figure 12A*.

**Source data 2.** Source data for statistical analysis of mEPSCs decay time in *Figure 12B*.

does not prompt this sharp decline in mEPSCs' decay, suggesting that the JCasp domain of APP is primarily involved in the presynaptic functions of APP.

## Discussion

Using an unbiased proteomic methodology and competition experiments, we found that the NH$_2$-terminal activation-induced cytidine deaminase (AID) region, JCasp, is the APP domain that binds presynaptic proteins regulating synaptic transmission. We have used a peptide encompassing this JCasp region to test whether the interaction of APP with proteins that regulate synaptic vesicles exocytosis enables APP to tune synaptic vesicles release. Strikingly, intracellular delivery of JCasp caused a strong reduction in basal synaptic transmission. Consistent with a presynaptic action, this reduction was associated with an increase in PPF and FF, as well as in mEPCS frequency but not amplitude, rise time, and decay time. These effects are evident in WT synapses but absent at APP-lacking synapses. In addition, Pen1-JCasp reduced the rate of synaptic vesicles depletion without affecting recycling. Taken together, these results indicate that JCasp acts as a dominant negative inhibitor of APP, and that APP facilitates the release of excitatory synaptic vesicles, likely via the interaction with and functional regulation of the molecular machinery that controls synaptic vesicles exocytosis.

The evidence that glutamatergic synaptic vesicles release is normal in the hippocampus of young *App*-KO mice suggests that compensatory mechanisms counterbalance the loss of APP function. Given the functional redundancy of APP and APLP2 in neuromuscular junctions patterning and survival (*Barbagallo et al., 2011b*, *Li et al., 2010*, *Wang et al., 2005*), we thought that APLP2 could compensate for loss of APP function at hippocampal synapse. Consistent with this hypothesis, we found that the intracellular domain of APLP2, ALID2, interacts with key regulators of synaptic vesicle exocytosis. A systematic characterization of SC–CA1 synaptic transmission revealed that APP and APLP2 have redundant functions in the hippocampus as indicated by the following observations: 1) *App*-KO animals showed normal synaptic transmission at SC–CA1 excitatory synapses; 2) *Aplp2*-KO mice presented a significant reduction in mEPSCs frequency; 3) *App/Aplp2*-dKO mice showed increased PPF and FF, which was not observed in single *App*-KO and *Aplp2*-KO mice, as well as reduced mEPSCs frequency.

Overall, the synaptic phenotype observed in *App/Aplp2*-dKO mice is very similar to that induced by Pen1-JCasp in WT animals, suggesting a primary functional role for the intracellular domains of APP and APLP2 in tuning glutamate release. A notable exception is represented by an increase in mEPSC decay time – present in *App/Aplp2*-dKO mice but not in hippocampal slices incubated with Pen1-JCasp. This observation suggests that Pen1-JCasp can unveil the role of the JCasp domain of APP without directly affecting the function of other APP region, such as, for example, Aβ or the APP extracellular domain.

Our results indicate that JCasp is an inhibitor of APP but not APLP2. This specificity of JCasp for APP allows to study the function of the APP intracellular domain in WT adult animals, eliminating the confounding effects associated with both constitutive and conditional *App*-KO models. In fact, it permits acute inhibition of the functions of this APP domain in WT adult animals, circumventing developmental issues and compensatory mechanisms, such as those mediated by APLP2 and/or APLP1. The results presented here indicate that APP and APLP2 facilitate the release of excitatory synaptic vesicles, likely via the interaction with and functional regulation of the molecular machinery that controls synaptic vesicle exocytosis. How exactly these presynaptic APP interactions tune vesicles exocytosis and how this function is regulated remain to be elucidated.

While a recent study in primary neuronal cells has shown that Aβ can increase P$r$ by enhancing presynaptic Ca$^{+2}$ influx (*Fogel et al., 2014*), our findings support a Ca$^{+2}$ influx-independent mechanism. These two mechanisms of action of APP are not mutually exclusive and could act synergistically to control neurotransmitter release. Inherited or acquired factors leading to altered APP metabolism and Aβ levels could negatively impact this physiological function of APP and, eventually, contribute to the pathogenesis of AD.

It is worth noting that AID/AICD and JCasp are natural proteolytic products of APP. The mechanisms by which cleavage of APP can regulate synaptic transmission are twofold; 1) severing the APP–presynaptic multi-molecular complex from membranes; and 2) liberating cytosolic proteolysis fragments of APP, such as AID/AICD and JCasp, which could act, as we show for Pen1-JCasp, as dominant negative inhibitors of APP. AID/AICD and JCasp have very short half-lives and are hardly detectable in vivo (*Cupers et al., 2001*, *Giliberto et al., 2010*, *2008*, *Passer et al., 2000*). It therefore appears that evolution has selected mechanisms, which are perhaps expensive in terms of

genetic information and energy usage, to tightly regulate the levels of AID/JCasp, suggesting that AID/AICD and JCasp are biologically active and need to be strictly controlled.

APP plays a central role in the pathogenesis of both sporadic AD and FAD. Many previous studies investigating synaptic dysfunction induced by Aβ, which derives from APP processing and is considered the main mediator of AD pathogenesis, have suggested that early pathogenic changes in AD were driven by postsynaptic impairments (*Cissé et al., 2011*, *Ferreira et al., 2012*, *Li et al., 2009*, *Mucke and Selkoe, 2012*, *Parameshwaran et al., 2008*, *Roberson et al., 2011*, *Shankar et al., 2007*, *Um et al., 2012*). However, the presynaptic function of APP described here , together with evidence that the other familial Alzheimer's proteins PS1 and PS2 regulate glutamate release in mature neurons by a presynaptic mechanism (*Fogel et al., 2014*, *Wu et al., 2013*, *Xia et al., 2015*, *Zhang et al., 2009*), support the possibility that deficits in neurotransmitter release represents a convergent mechanism involved in AD.

## Materials and methods

### Mice and ethics statement

Mice were handled according to the Ethical Guidelines for Treatment of Laboratory Animals of the NIH. The procedures were described and approved by the Institutional Animal Care and Use Committee (IACUC). The *App*-KO (Stock No: 004133) and *Aplp2*-KO (Stock No: 004142) mice were originally purchased from The Jackson Laboratory on a C57BL/6J background. Mice were crossed to obtain $App^{+/-}/Aplp2^{+/-}$ animals. $App^{+/-}/Aplp2^{+/-}$ mice were crossed to 129 mice to select $App^{+/-}/Aplp2^{+/-}$ animals on a C57BL/6J/129 mixed background. $App^{+/-}/Aplp2^{+/-}$ animals on a C57BL/6J/129 were crossed and WT, *App*-KO, *Aplp2*-KO, or *App/Aplp2*-dKO were used for experiments. C57BL/6J mice were generated at the Einstein animal facility.

### Mouse brain and hippocampal preparation

The brain samples used for proteomic experiments were prepared as follows. Brains were homogenized in 5 mM (4-(2-hydroxyethyl)-1-piperazineethanesulfonic acid (HEPES)/NaOH pH 7.4, 1 mM ethylenediaminetetraacetic acid (EDTA), 1 mM ethylene glycol tetraacetic acid (EGTA), 0.25 M sucrose supplemented with protease and phosphatase inhibitors (Thermo Scientific). Brain homogenates were centrifuged at 800$g$ for 10 min. Supernatant was collected and centrifuged at 9200$g$ for 10 min to obtain the pellet (P2) and the supernatant (S2) fractions. The P2 fraction, which contains synaptosomes, was solubilized in 0.32 M sucrose, 20 mM Tris-base pH 7.4, 1 mM EDTA, incubated for 30 min at 4°C and centrifuged at 21000$g$ for 15 min. The supernatant (LS1) was collected. The S2 and LS1 fractions were combined and used for the proteomic experiments. For Western blot analysis on hippocampal slices treated with either Pen1-JCasp or Pen1-ScJCasp (*Figure 5*), P2 fractions were prepared and solubilized in 1% TritonX100 RIPA buffer for 30 min at 4°C and centrifuged at 21000$g$. The triton soluble supernatant fractions were separated on a 16.5% Tris-Tricine sodium dodecyl sulfate polyacrylamide gel electrophoresis (SDS-PAGE), to obtain a better separation of APP-CTF species, and transferred onto nitrocellulose membranes. For Western blot analysis of pull-downs pre-incubated with JCasp or ScJCasp (*Figure 1D*), samples were separated on a 4–20% SDS-PAGE and transferred onto nitrocellulose membranes. The following antibodies were used in Western blots: anti-APP C-terminal AbD (Zymed), anti-APP N-terminal 22C11 (Millipore), anti-Actin (Cell Signaling), anti-Syp (Synaptic Systems, Goettingen, Germany), anti-Vamp2 (Synaptic Systems), anti-Syt2 (Synaptic Systems) and anti-Ddb11 (Cell Signaling).

### Pull-down assays with StrepTag peptides and mass spectrometry

The StrepTag peptides were immobilized on StrepTactin column (IBA, St. Louis, MO), incubated with StrepTactin pre-cleared WT brain S2+LS1 fractions, and washed and eluted with desthiobiotin. The following analysis was performed by MSBioworks (Ann Arbor, Michigan). To prepare samples for mass spectrometry, the volume of each sample was reduced to 50 µL by vacuum centrifugation, 20 µL of each concentrated sample was processed by SDS-PAGE using a 10% Bis-Tris NuPAGE gel (Invitrogen) with the 2-(*N*-morpholino)ethanesulfonic acid (MES) buffer system, the gel was run approximately 2 cm. The mobility region was excised into 10 equal sized segments and in-gel digestion was performed on each using a robot (ProGest, DigiLab) with the following protocol: washed

with 25 mM ammonium bicarbonate followed by acetonitrile. Reduced with 10 mM dithiothreitol at 60°C followed by alkylation with 50 mM iodoacetamide at RT. Digested with trypsin (Promega) at 37°C for 4 hr. Quenched with formic acid and the supernatant was analyzed directly without further processing. Each digest was analyzed by nano liquid chromatography–mass spectrometry (LC/MS/MS) with a Waters NanoAcquity High Performance Liquid Chromatography (HPLC) system interfaced to a Thermo Fisher Q Exactive mass spectrometer. Peptides were loaded on to a trapping column and eluted over a 75 µm analytical column at 350 nL/min; both columns were packed with Jupiter Proteo resin (Phenomenex). The mass spectrometer was operated in data-dependent mode, with MS and MS/MS performed in the Orbitrap at 70,000 and 17,500 full width at half maximum (FWHM) resolution, respectively. The 15 most abundant ions were selected for MS/MS.

## Immunofluorescence

The mice were anesthetized by intraperitoneal injection (IP) of Avertin and perfused with ~20–30 ml of Soresan's Buffer followed by 30–50 ml of 4% paraformaldehyde (PFA) fixative. After the perfusion brains were quickly removed and fixed in 4% PFA in Soresan's Buffer overnight at 4°C. Brains were sectioned (50 µm thick) with a vibratome; slices were washed with phosphate buffer saline (PBS) and incubated for 90 min with pre-blocking agent containing 10% normal goat serum (NGS) in PBS, 0.02% saponin, and 0.1% Triton. Slides were washed in PBS and incubated overnight with primary antibodies Neun (1:300, Millipore), APP Y188 (1:200, Abcam), and Snap25 (1:300, Synaptic Systems) prepared in pre-blocking agent solution. After washes, sections were incubated with secondary Alexa 488 and 594 fluorescent antibodies (Molecular Probes) at room temperature for 2.5 hr. After washes, slices were fixed on glass slides and incubated with the antifade (Prolong Gold). Images were captured using SP5 confocal microscope (Leica), under 10X and 63X magnification.

## Electrophysiology on brain slices

We used C57BL/6J WT mice aged 3–12 weeks of both sex (experiments shown in *Figures 3*, *4*, *5*, *7*, *8*), as well as 16–28-day-old C57BL/6J/129 hybrid mice, which were either WT, *App*-KO, *Aplp2*-KO, or *App/Aplp2*-dKO (experiments shown in *Figures 9*, *11, 12*). The experimenter was blind to genotype and/or treatment. Coronal brain slices containing the hippocampal formation were prepared as previously described (*Talani et al., 2011*). In brief, animals were anesthetized with isoflurane, following the University of California San Diego Institutional Animal Care and Use Committee (IACUC) approved protocol. Brains were rapidly removed from the skull and placed in an ice-cold modified ACSF solution containing (in mM): 215 sucrose, 2.5 KCl, 1.6 $NaH_2PO_4$, 4 $MgSO_4$, 1 $CaCl_2$, 4 $MgCl_2$, 20 glucose, 26 $NaHCO_3$ (pH 7.4 equilibrated with 95% $O_2$ and 5% $CO_2$). Coronal brain slices (400 µm thick) were prepared with a Vibratome VT1200S (Leica Microsystems, Germany) and then incubated at room temperature in a physiologic ACSF, containing (in mM): 120 NaCl, 3.3 KCl, 1.2 $Na_2HPO_4$, 26 $NaHCO_3$, 1.3 $MgSO_4$, 1.8 $CaCl_2$, 11 Glucose (pH 7.4 equilibrated with 95% $O_2$ and 5% $CO_2$). The hemi-slices were transferred to a recording chamber perfused with ACSF at a flow rate of ~2 ml/min using a peristaltic pump; experiments were performed at 28.0 ± 0.1°C. All recordings were performed using a multiclamp 700B amplifier (Molecular Devices). For extracellular field recordings (fEPSP recordings), a patch-type pipette was fabricated on a micropipette puller (Sutter Instruments, Novato, CA, USA), filled with 1M NaCl, and placed in the middle third of *stratum radiatum* in area CA1. Field excitatory postsynaptic potentials (fEPSPs) were evoked by activating SCs with a patch-type pipette (monopolar stimulation) filled with ACSF and placed in the middle third of *stratum radiatum* 150–200 µm away from the recording pipette and at approximately the same slice depth (150–200 µm). Square-wave current pulses (60 µs pulse width) were generated by a stimulus generator (Master 8, AMPI) and delivered through a stimulus isolator (Isoflex, AMPI). I/O function was generated by stimulating in 0.1 mA steps from 0.1 to 1.0/1.2 mA. Paired-pulse ratio (PPR) was measured by delivering two stimuli at 20, 40, 80, 200, 500, and 1000 ms inter-stimulus intervals. Synaptic facilitation was examined by repetitive stimulation (10 stimuli) at 1, 5, 10, and 20 Hz. Depletion of synaptic transmission was elicited by a long train (500 stimuli, 28 Hz) in 5 mM extracellular $Ca^{2+}$ and in the presence of the NMDA receptor antagonist 2-amino-5-phosphopenta-noic acid (DL-AP5) (100 µM). Recovery was assessed with nine stimuli at 2 Hz. Depletion and recovery plots were generated by normalizing responses to the peak amplitude of the fEPSP in the depletion train. The stimulus strength was adjusted so that the basal synaptic amplitude was similar

(e.g., 0.5–1.0 mV) across experiments. To monitor miniature excitatory postsynaptic currents (mEPSCs), patch-clamp whole-cell recordings were performed from CA1 pyramidal cells, and we used an infrared differential interference contrast microscope (Nikon eclipse E600FN, Morrel Instrument Company Inc, Melville New York). The cells were voltage-clamped at -65 mV, and the patch pipette (resistance 3–6 MΩ) was filled with an internal solution containing (in mM): 131 CsCl, 8 NaCl, 1 $CaCl_2$, 10 EGTA, 10 Glucose, 10 HEPES (pH = 7.3, 286 mmol/Kg). Series resistance (5 and 15 MΩ) was constantly monitored throughout the experiments by delivering a -5 mV, 80 ms voltage step, and cells with >10% change in series resistance were excluded from analysis. mEPSCs were recorded in the presence of 1 µM TTX and 15 µM 1(S), 9(R)(−)bicuculline methiodide, to block action potentials and GABA-A receptors, respectively. To calculate average mEPSCs recorded from at least three (*Figures 4* and *8*) and five (*Figure 12*) CA1 pyramidal neurons (between ~1000 and 600 events each) were normalized, aligned, and averaged.

*Data acquisition and statistical analysis.* Output signals were acquired at 5 kHz, filtered at 2.4 kHz, and stored online using pCLAMP 10.3 Electrophysiology Data Acquisition and Analysis Software (Molecular Devices). mEPSCs were analyzed using Mini Analysis software (Synaptosoft). In all cases the experimenter was blind to genotype and/or treatment. Data are presented as means ± SEM. Statistical comparisons of pooled data were performed by unpaired t-test or ANOVA (one- or two-way) using Prism software (GraphPad, San Diego,CA, USA). A *p* value of <0.05 was considered statistically significant.

Drugs were obtained from Sigma-Aldrich 1(S), 9(R)(−)bicuculline methiodide, Biotium (TTX) and Enzo Life Science (DL-AP5). Stock solutions were prepared in water and dimethyl sulfoxide (DMSO) and added to the ACSF as needed.

---

## Additional information

### Funding

| Funder | Grant reference number | Author |
| --- | --- | --- |
| Alzheimer's Association | Zen-11-201425 | Tomas Fanutza<br>Dolores Del Prete<br>Luciano D'Adamio |
| National Institute on Aging | 1RO1AG041531,<br>1RO1AG033007,<br>5R21AG048971 | Tomas Fanutza<br>Dolores Del Prete<br>Luciano D'Adamio |
| BrightFocus Foundation | A2013046F | Dolores Del Prete |
| National Institute on Drug Abuse | DA017392 | Pablo E Castillo |
| National Institute of Mental Health | MH081935 | Pablo E Castillo |

The funders had no role in study design, data collection and interpretation, or the decision to submit the work for publication.

### Author contributions

TF, Prepared samples for electrophysiology recordings. Performed the electrophysiology recordings. Analyzed and interpreted the electrophysiology recordings, Acquisition of data, Analysis and interpretation of data; DDP, Performed biochemical experiments. Generated the KO mice. Performed IF experiments. Prepared samples for electrophysiology recordings. Analyzed and interpreted the electrophysiology recordings, Acquisition of data, Analysis and interpretation of data; MJF, Performed the proteomic analysis; PEC, Contributed to the analysis, interpretation and planning of electrophysiology experiments. Contributed to writing the paper. Analysis and interpretation of data, Drafting or revising the article; LD'A, Designed experiments. Performed biochemical experiments. Generated the KO mice. Analyzed and interpreted the data. Wrote the manuscript; Conception and design, Acquisition of data, Analysis and interpretation of data; Drafting or revising the article

## Ethics

Animal experimentation: This study was performed in strict accordance with the recommendations in the Guide for the Care and Use of Laboratory Animals of the National Institutes of Health. All of the animals were handled according to approved institutional animal care and use committee (IACUC) at the Albert Einstein College of Medicine in animal protocol number 20130509. All surgery was performed under sodium pentobarbital anesthesia, and every effort was made to minimize suffering.

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
