## [Decision Letter]

Thank you for submitting your work entitled "APP interacts with the synaptic release machinery and facilitates transmitter release at hippocampal synapses" for peer review at *eLife*. Your submission has been favorably evaluated by a Senior editor and three reviewers, one of whom is a member of our Board of Reviewing Editors. One of the three peer reviewers, Jie Shen, has agreed to reveal his identity.

The reviewers have discussed the reviews with one another and the Reviewing editor has drafted this decision to help you prepare a revised submission.

Summary:

This paper by Fanutza et al. focuses on the functional significance of the N-terminal part of the intracellular APP domain (JCasp) in synaptic transmission and short-term plasticity in CA3-CA1 synapses. This work builds further on their previous study (Del Prete et al., 2014). They first show specific interaction of a series of synaptic vesicle associated proteins with the membrane proximal part of the APP intracellular domain (JCasp). They then show that this fragment, when modified with penetratin inhibits basal synaptic transmission and enhances short-term synaptic facilitation in CA3-CA1 synapses. Moreover, the authors found that JCasp imposes a tonic inhibition on spontaneous vesicle release, decreasing the mEPSC frequency. This study addresses an important question in neurobiology, i.e. the physiological impact of the intracellular APP domain in Schaffer collateral presynaptic function. The manuscript further explores the interaction between the intracellular domain of APP and proteins involved in neurotransmitter release. Specifically, the authors showed that JCasp, a fragment that is the N-terminus part of the APP intracellular domain and is cleaved by g-secretase and caspase, is involved in presynaptic short-term plasticity and neurotransmitter release. Overall, the paper provides much needed mechanistic insight into normal function of APP at presynaptic terminals where it is abundantly localized.

Essential revisions:

1) The quality of the immunofluorescent data needs to be upgraded. It is not possible to decide whether the APP staining is cellular or in synapses.

2) The whole story critically depends on one type of experiment: the effect of penetratin-coupled Pen1-JCasp on synaptic transmission. The authors provide a series of important controls on the experiment (penetratin deficient, scrambled peptide…), but no independent experimental approach to confirm their hypothesis that APP is involved in neurotransmission in the way they propose. The fact that the APP KO neurons do not show the same phenotype as the neurons treated with Pen1-JCasp is worrying. I can see that compensation by APLP2 could provide an explanation, but it would be great to actually demonstrate this and to show that APP, but not APP in which the JCasp part is deleted can rescue this in double knock out neurons. Can APPL2 overexpression rescue the phenotype induced by penetratin-JCasp? Alternatively does overexpression of the identified JCasp interacting proteins rescue the penetratin-JCasp mediated effect? Or have the authors other ideas to confirm their working hypothesis? Not all of these experiments have to be performed but some work along these lines could strengthen considerably the claims of the paper.

3) The authors should present the evidence for Pen1-JCasp penetration to the intracellular compartments of neurons in slices after 5 hr of pre-incubation. All the experiments were done following 5 hr pre-incubation of slices with the peptides. Was it 5 hr after the recovery of slices in large ACSF volume? Given that the experiments take at least 1 hr, all the measurements in the paper were done in late slices with modified properties. I wonder whether the authors find no difference in synaptic facilitation between WT and APP-KOs also in fresh slices (1-2 hr after recovery). I think it is important to repeat the major finding – increase in synaptic facilitation – after ICV injection of Pen1-JCasp and preparing the slices several hours after it.

4) Figure 7: mEPSC amplitude – something is wrong with the scale bar and the average amplitude of 3 pA. What is the noise level in the system? In Figure 3, the average amplitude is ~7 pA. The authors should show cumulative distributions for all the mEPSC measurements in the paper and explain in Methods clearly the detection threshold and which filters were applied for mEPSC recording/analysis. Why is the rise time ~40% larger in Figure 7 in comparison to Figure 3?

5) In the Figure 7, the authors show that the effect of JCasp on miniature spontaneous events persists in the presence of cadmium, a non-selective blocker of voltage-gated calcium channels. Based on these results, they concluded that JCasp generally reduces synaptic vesicle release in a Ca-independent manner. As spontaneous and evoked synaptic vesicle release may be differentially regulated, the authors should test the effect of JCasp on presynaptic Ca transients evoked by action potentials or, at least, discuss this caveat in the text and carefully interpret the mEPSC data. Discussion on the mechanism of Ca-independent, JCasp-mediated tonic inhibition would be helpful – could the authors suggest the specific stage in the exocytosis that is under a tonic block of JCasp.

6) The "Appresyome" is a vague description to indicate presynaptic proteins interacting with APP. I would not use this terminology, it is imprecise. What proteins are precisely responsible for the effects? The authors remain also very vague whether these interactions occur at the level of the synaptic vesicle or at the level of the presynaptic membrane. Is APP part of the snare complexes?

---

## [Author Response]

*Essential revisions:*

*1) The quality of the immunofluorescent data needs to be upgraded. It is not possible to decide whether the APP staining is cellular or in synapses.*

We apologize for not having clearly stated the rationale for the IF experiments. The purpose of the experiment shown in Figure 1 of the original manuscript (which is now Figure 1 of the revised version) was to show the distribution of endogenous APP in the hippocampus. Albeit this may seem as a superfluous exercise, we believe it is important in view of the recent report form Hui Zheng’s laboratory (Guo, Li et al., 2012) showing that most of the antibodies that have been used in the past to determine the localization of APP were non-specific (i.e. they stained *App*-KO and WT material in a similar fashion). The authors established with clarity that, among those tested, the Y188 antibody was the only specific antibody. We are unaware of any publication showing the expression of endogenous APP in mouse hippocampus using the Y188 antibody or other antibodies with appropriate controls (i.e. *App*-KO tissues to determine specificity). All other information concerning APP distribution in the hippocampus is derived from analysis of mice overexpressing APP. It is worth noting that: a) the distribution of overexpressed APP does not necessarily reflect the “normal” distribution of endogenous APP; b) detecting endogenous APP is objectively a much more difficult task then detecting overexpressed APP.

The staining with Y188 (the specificity of Y188 was confirmed by the evidence that this antibody does not stain *App*-KO hippocampi) showed high level of APP staining in cell bodies of DG, CA3 and CA1 neurons. On the contrary, the staining in other regions, such as the stratum radiatum containing the CA3 projections to CA1 neurons, is more diffused and less intense. This is not surprising since cell bodies are where the ER and Golgi structure are highly concentrated: this is where APP, a type I membrane protein, is synthesized and undergoes maturation.

From the cell bodies, APP will be transported along dendrites at synapses. Thus, it is expected to find that APP in more concentrated in cellular bodies as compared to dendrites and synapses. We have now added a staining in Figure 1 showing that APP is found in the stratum radiatum, where it partially co-localizes with the specific presynaptic marker Bassoon, supporting a presynaptic function of APP in the SC pathway. This staining is specific since it is absent in *App*-KO hippocampi.

Technical issues could also explain the quantitative differences in Y188 staining between CA1/CA3 cell bodies and stratum radiatum. Y188 recognize the C-terminal epitope. This epitope may be functionally engaged in multi molecular interactions (this is actually our working model) in dendrites/synapses -perhaps immature APP that is in cell body is much less BUSY interacting- therefore it may be scarcely accessible to Y188.

Overall, our findings are consistent with previous observations showing a pre-synaptic localization of APP (Del Prete et al., 2014, Fogel, Frere et al., 2014, Gautam et al., 2015, Kohli, Pflieger et al., 2012, Lassek, Weingarten et al., 2014, Lassek et al., 2013, Lundgren et al., 2015, Norstrom et al., 2010).

*2) The whole story critically depends on one type of experiment: the effect of penetratin-coupled Pen1-JCasp on synaptic transmission. The authors provide a series of important controls on the experiment (penetratin deficient, scrambled peptide...), but no independent experimental approach to confirm their hypothesis that APP is involved in neurotransmission in the way they propose. The fact that the APP KO neurons do not show the same phenotype as the neurons treated with Pen1-JCasp is worrying. I can see that compensation by APLP2 could provide an explanation, but it would be great to actually demonstrate this and to show that APP, but not APP in which the JCasp part is deleted can rescue this in double knock out neurons. Can APPL2 overexpression rescue the phenotype induced by penetratin-JCasp? Alternatively does overexpression of the identified JCasp interacting proteins rescue the penetratin-JCasp mediated effect? Or have the authors other ideas to confirm their working hypothesis? Not all of these experiments have to be performed but some work along these lines could strengthen considerably the claims of the paper.*

The fact that *App*-KO hippocampi of young mice showed normal synaptic transmission is expected. Although *App*-KO mice have been available since 1992, deficits in PPF and FF in the SC pathway have never been reported in these animals. In addition, deficits in long-term plasticity (long-term potentiation, LTP) are obvious only in aged, but not in juvenile, *App*-KO mice [reviewed in (Korte, Herrmann et al., 2012)]. As well pointed out by the reviewers, APP and APLP2 have redundant functions; APLP2 compensates for APP in *App*-KO mice and, vice versa, APP compensates for APLP2 in *Aplp2*-KO mice. *App/Aplp2*-dKO mice have dramatic deficits in NMJ patterning; yet *App*-KO and *Aplp2*-KO mice have absolutely normal NMJ development. More vividly, *App/Aplp2*-dKO mice have a very short lifespan (~30 days); yet *App*-KO and *Aplp2*-KO mice have substantially normal lifespans. These observations demonstrate that *App/Aplp2*-dKO mice have extremely striking phenotypes, whereas single KO mice are essentially normal, clearly illustrating the power of the compensatory mechanisms between APP and APLP2.

We thought that the data obtained with JCasp are important and could stand alone mainly for two reasons: 1) the evidence that Pen1-JCasp can acutely inhibit the normal function of APP in young wild type mice can be considered a strength, since it is a great tool to unveil the functional role of APP (and more specifically of a specific APP domain, the intracellular domain) avoiding compensatory mechanisms and/or the ablation of functions associated with other domains of APP; 2) given that JCasp is a naturally occurring APP fragment, understanding its function is intrinsically important, just like studying the function of other APP metabolites including Aβ, AID/AICD, the recently characterized Aη-α, sAPPβ, sAPPα, β-CTF, α-CTF etc.

Having established that JCasp is functionally active, we went one step further and tested JCasp in *App*-KO hippocampi to better understand the mechanism of action of JCasp. We felt that experiments addressing compensation issues may have sidetracked the reader and diluted the main messages of the paper.

Yet, it is of value to determine whether APP and APLP2 share common synaptic functions and whether APLP2 can compensate the loss of APP function in *App*-KO hippocampi. In fact, we have been testing this hypothesis.

First, we have used a proteomic approach to characterize the brain interactome of the APLP2 intracellular domain. The rational is that if the interaction of APP with key regulators of exocytosis is important for the described synaptic function of APP, and if APLP2 compensates the loss of this function in *App*-KO, then the intracellular domain of APLP2 should also interact with key regulators of exocytosis. The data are now included in Table 2 and show that also the APLP2 intracellular domain interacts with pre-synaptic proteins, including of Vamp2, Syp and Syt2.

Second, we have tested whether JCasp interferes with these interactions, since the electrophysiological data suggest that Pen1-JCasp is a dominant negative of APP but not APLP2 (Pen1-JCasp does not alter synaptic transmission in *App*-KO mice, which still express APLP2). As shown in Figure 10, JCasp does not alter the interaction of Vamp2, Syp and Syt2 with ALID2.

Then, we have tested synaptic transmission in WT, *App*-KO, *Aplp2*-KO and *App/Aplp2*-dKO mice derived from *App^KO/WT^/Aplp2^KO/WT^* breeding pairs. These experiments, that are shown in Figure 11 and Figure 12 and were underway when the manuscript was originally submitted, indicate that: 1) APP and APLP2 facilitate glutamate release; 2) APLP2 compensates for loss of APP function in *App*-KO mice and, vice versa, that APP compensates for loss of APLP2 function in *Aplp2*-KO mice. These experiments are described in a new sub-chapter of the Results section entitled “Loss of APP and APLP2 reduces excitatory synaptic transmission”.

*3) The authors should present the evidence for Pen1-JCasp penetration to the intracellular compartments of neurons in slices after 5 hr of pre-incubation.*

To visualize intracellular peptides, we have synthesized JCasp and Pen1-JCasp linked to Biotin at the NH_2_-terminus. As we show in Figure 2 of this revised manuscript, Pen1-JCasp, but not JCasp, penetrates into the intracellular compartments of neurons. The results are discussed in the section entitled “Intracellular delivery of JCasp decreases excitatory synaptic transmission”, second paragraph.

*All the experiments were done following 5 hr pre-incubation of slices with the peptides. Was it 5 hr after the recovery of slices in large ACSF volume? Given that the experiments take at least 1 hr, all the measurements in the paper were done in late slices with modified properties. I wonder whether the authors find no difference in synaptic facilitation between WT and APP-KOs also in fresh slices (1-2 hr after recovery). I think it is important to repeat the major finding – increase in synaptic facilitation – after ICV injection of Pen1-JCasp and preparing the slices several hours after it.*

We apologize for not having explained more clearly the design of our experiments. In this revised version, we specify (in the section “Intracellular delivery of JCasp decreases excitatory synaptic transmission”): “The experiments were performed as follows: soon after preparation, the brain slices were incubated in 200 ml of ACSF containing the indicated concentrations of either Pen1-JCasp or the negative control peptide Pen1-ScJCasp. Recordings were performed between ~5 and 8 hr after preparation of slices and incubation with Pen1-peptides.”

As suggested by the reviewers, to further stress the point that the results were not artificially affected by the experimental design, we compared facilitation on fresh *vs*. old slices in wild type and *App*-KO mice. Normally, we allow the brain slices to recover for 3 hr after cutting. For this experiment, we have defined as “fresh slices” those analyzed between 3 and 5 hr post-cutting, and as “old slices” those analyzed between 5 and 8 hr post-cutting. As shown in Figure 13, synaptic facilitation is not significantly affected within this time frame in neither genotype, and there is no significant difference between wild type and *App*-KO recordings both in “fresh” and in “old” slices. Moreover, a new set of experiments conducted on WT, *App*-KO, *Aplp2*-KO and *App/Aplp2*-dKO (and discussed above) confirms that, at least at 3-4 weeks of age, *App*-KO mice do not show deficits in facilitation.

Author response image 1.The “age” of the hippocampal slices does not affect facilitations in both WT and App-KO SC:CA1 synapses.Frequency facilitation was tested at 20 Hz.**DOI:**
http://dx.doi.org/10.7554/eLife.09743.038

In addition, if Pen1-JCasp were affecting synaptic transmission by altering neurons’ recovery and not by altering APP functions, Pen1-JCasp would probably have a similar effect in WT and *App*-KO hippocampal neurons. In view of these clarifications and new data, we hope the reviewer will agree in that testing synaptic facilitation in slices prepared from mice after ICV injection of peptides does not seem to be required. ICV injection experiments – especially when using peptides that have never been tested before – are not trivial to set up. The effects of Pen1-JCasp on synaptic facilitation are highly dependent on the concentration of the peptide (a clear example is illustrated in Figure 3 of the paper). Determining the delivery conditions (peptide concentration in the solution being injected/volume injected/frequency of injection/location of delivery etc.) needed to achieve working concentrations of Pen1-JCasp and Pen1-ScJCasp in the hippocampus of a live animal is not straightforward.

4) Figure 7: mEPSC amplitude – something is wrong with the scale bar and the average amplitude of 3 pA. What is the noise level in the system? In Figure 3, the average amplitude is ~7 pA. The authors should show cumulative distributions for all the mEPSC measurements in the paper and explain in Methods clearly the detection threshold and which filters were applied for mEPSC recording/analysis. Why is the rise time ~40% larger in Figure 7 in comparison to Figure 3?

We thank the reviewers for pointing out these errors. We confirm that there was a mistake not only in the calibration bars but a digital filter was mistakenly applied to the raw data. Moreover, while the experiments shown in the old Figure 3 were performed in 4 weeks old C57Bl6 mice, the experiments in Figure 8 were performed in p18/20 C57Bl6/129 mixed background mice. The recording conditions and the different genetic background may contribute to the differences between experiments that are noted by the reviewer and the low amplitude observed in the experiment reported in Figure 8. We have repeated these experiments in hippocampal slices prepared from 3/4 weeks old C57Bl6/J mice, and removed the digital filter. The results are now shown in the revised Figure 4 and 8. In response to the reviewers we have also included cumulative probability curves. These new experiments not only confirm that the Pen1-JCasp reduces the frequency of mEPSCs but also confirm that Cd^+2^ does not alter the effects of Pen1-JCasp.

5) In the Figure 7, the authors show that the effect of JCasp on miniature spontaneous events persists in the presence of cadmium, a non-selective blocker of voltage-gated calcium channels. Based on these results, they concluded that JCasp generally reduces synaptic vesicle release in a Ca-independent manner. As spontaneous and evoked synaptic vesicle release may be differentially regulated, the authors should test the effect of JCasp on presynaptic Ca transients evoked by action potentials or, at least, discuss this caveat in the text and carefully interpret the mEPSC data. Discussion on the mechanism of Ca-independent, JCasp-mediated tonic inhibition would be helpful – could the authors suggest the specific stage in the exocytosis that is under a tonic block of JCasp.

Our experiments using the voltage-gated calcium channel blocker cadmium show that these channels do not play a significant role in the JCasp-mediated suppression of synaptic transmission. We therefore conclude that JCasp reduces glutamate release in a “calcium influx-independent manner” (Discussion). The fact that JCasp was equally effective in the presence of cadmium, strongly suggests that JCasp mainly acts downstream calcium influx. We do not claim that JCasp acts in a calcium-independent manner, and we have no evidence that JCasp differentially targets evoked vs. spontaneous release.

*6) The "Appresyome" is a vague description to indicate presynaptic proteins interacting with APP. I would not use this terminology, it is imprecise. What proteins are precisely responsible for the effects? The authors remain also very vague whether these interactions occur at the level of the synaptic vesicle or at the level of the presynaptic membrane. Is APP part of the snare complexes?*

In response to the reviewer’s suggestion, we have removed the term Appresyome. Defining which interaction (or interactions) is (are) “directly” responsible for the effects, whether synaptic vesicle APP and/or presynaptic membrane APP are involved, and whether APP is an integral part of the SNARE complex are extremely important. However, addressing these questions will likely require multiple approaches and several years of experimentation. As we discuss in our manuscript (in the section “Intracellular delivery of JCasp decreases excitatory synaptic transmission”): “This presynaptic APP-interacting network can stem from APP molecules localized either at synaptic vesicles or at presynaptic membranes and is almost certainly formed via the direct interaction of the intracellular domain of APP with some release machinery protein”. We also acknowledge (in the Discussion): “How exactly these presynaptic APP interactions tune vesicles exocytosis and how this function is regulated remain to be elucidated.”